# Molecular basis of β-lactam antibiotic resistance of ESKAPE bacterium *E. faecium* Penicillin Binding Protein PBP5

Yamanappa Hunashal[1], Ganesan Senthil Kumar [1,2], Meng S. Choy[1], Éverton D. D'Andréa[3], Andre Da Silva Santiago [4], Marta V. Schoenle[3], Charlene Desbonnet[5], Michel Arthur [6], Louis B. Rice[5], Rebecca Page[4] & Wolfgang Peti [1] ✉

Penicillin-binding proteins (PBPs) are essential for the formation of the bacterial cell wall. They are also the targets of β-lactam antibiotics. In *Enterococcus faecium*, high levels of resistance to β-lactams are associated with the expression of PBP5, with higher levels of resistance associated with distinct PBP5 variants. To define the molecular mechanism of PBP5-mediated resistance we leveraged biomolecular NMR spectroscopy of PBP5 – due to its size (>70 kDa) a challenging NMR target. Our data show that resistant PBP5 variants show significantly increased dynamics either alone or upon formation of the acyl-enzyme inhibitor complex. Furthermore, these variants also exhibit increased acyl-enzyme hydrolysis. Thus, reducing sidechain bulkiness and expanding surface loops results in increased dynamics that facilitates acyl-enzyme hydrolysis and, via increased β-lactam antibiotic turnover, facilitates β-lactam resistance. Together, these data provide the molecular basis of resistance of clinical *E. faecium* PBP5 variants, results that are likely applicable to the PBP family.

*Enterococci faecium*, an ESKAPE pathogen, causes severe and often fatal nosocomial and community-acquired infections[1–3]. Effective therapies are hindered by the increased resistance to many classes of antibiotics, especially β-lactams[4–6]. These antibiotics target penicillin-binding proteins (PBPs) that are part of a group of enzymes that form the peptidoglycan layer, a critical component of the bacterial cell wall that is essential for cell survival[7,8]. PBPs have either transpeptidase, carboxypeptidase, or endopeptidase activity that synthesize new and remodel existing peptidoglycan[9]. PBPs are classified by their enzymatic activity: (a) class A, bifunctional PBPs with both glycosyltransferase and transpeptidase activities; (b) class B, transpeptidases; and (c) class C, carboxypeptidases and endopeptidases[9].

In *enterococci*, reduced susceptibility to β-lactam antibiotics, including penicillin and anti-MRSA cephalosporin (or ceftaroline), results from the expression of a single, so-called *low affinity*, class B PBP designated PBP5 in *E. faecium*[10,11]. Related PBPs confer β-lactam resistance in *E. faecalis* (PBP4) and *Staphylococcus aureus* (PBP2a)[12]. Class B PBPs have four distinct, highly interconnected domains: (a) N1, which links the membrane-inserting helix with the transpeptidase (TP) domain; (b) N2, which has an unknown function; (c) nPB, which links N1 and N2 with the TP domain and (d) TP domain, which includes the active site (Fig. 1A, B, Supplementary Fig. 1A, B)[13]. The nucleophilic serine (S422) is located at the N-terminus of helix α2, while the oxanion hole is defined by the backbone nitrogen atoms of S422 and

[1]Department of Molecular Biology and Biophysics, University of Connecticut Health Center, Farmington, CT, USA. [2]National Institute of Immunology, New Delhi, India. [3]Department of Chemistry and Biochemistry, University of Arizona, Tucson, AZ, USA. [4]Department of Cell Biology, University of Connecticut Health Center, Farmington, CT, USA. [5]Department of Medicine, Rhode Island Hospital, Warren Alpert Medical School of Brown University, Providence, RI, USA. [6]INSERM, Sorbonne Université, Université Paris Cité, Paris, France. ✉e-mail: peti@uchc.edu

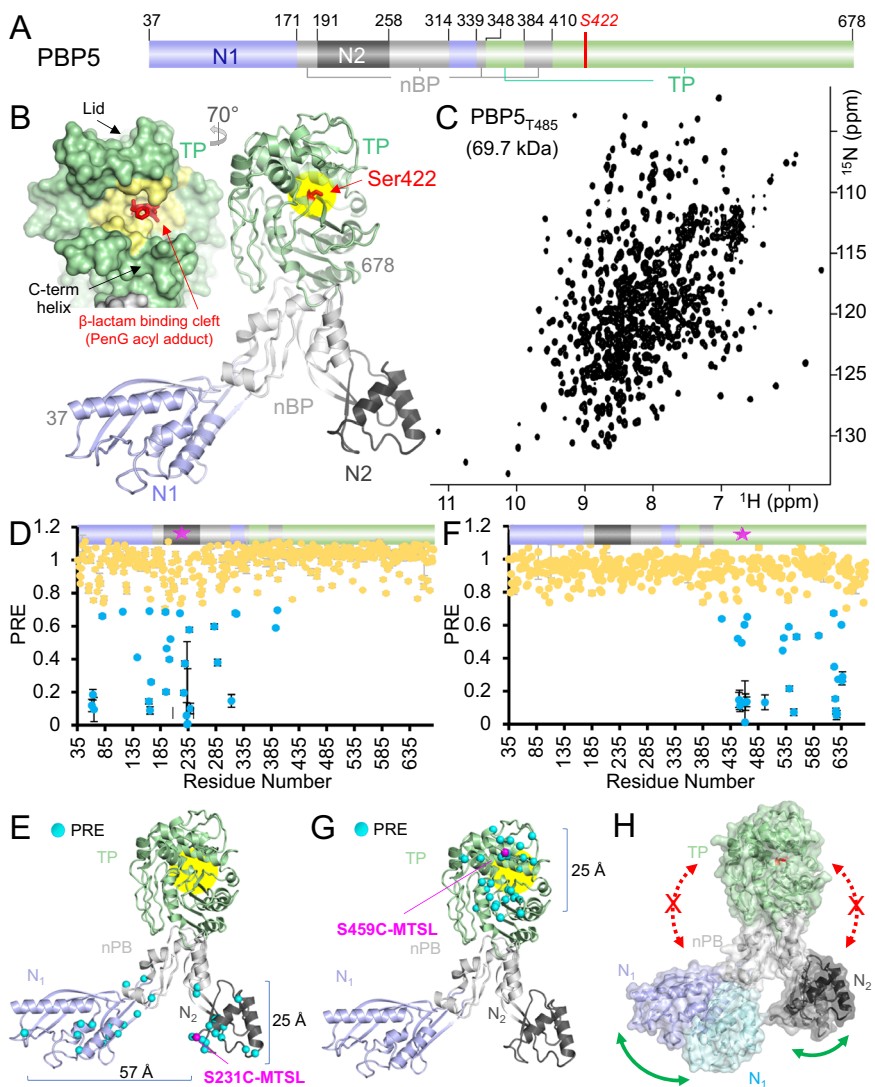

**Fig. 1 | PBP5 domain dynamics in solution. A** Domain structure of PBP5, colored according to its four distinct domains: N-terminal domain 1 (N1), blue; N-terminal domain 2 (N2), dark gray; non-penicillin binding domain (nPB), light gray; trans-peptidase domain (TP), green. Residues numbers at domain transitions are shown. **B** Crystal structure of PBP5 (PDBid 6MKA), colored by domains as described in A. The N- (residue 37) and C-terminus (residue 678) are labeled, and the catalytic serine (S422) is shown as red sticks and labeled. *Inset*, surface representation of the TP domain as the acyl-adduct with penG (PDBid 6MKG). PenG is shown as red sticks. PBP5 residues within 5 Å of penG are colored yellow, highlighting the β-lactam binding groove. This groove is surrounded by the PBP5 'lid' (aa 443–471) and C-terminal helix (aa 655–671), as labeled. **C** 2D [¹H,¹⁵N] TROSY spectrum of PBP5 (refolded). **D** Relative intensity plot of PBP5 S231C-MTSL paramagnetic relaxation enhancement (PRE) vs. residue number. Upper bar colored by the PBP5 domain;

residues with reduced PRE values are colored cyan. **E** PBP5 S231C residues with reduced PREs are plotted on the structure PBP5 (PDBid: 6MKA) as cyan balls. Static distances between domains indicated. **F** Relative intensity plot of PBP5 S459C-MTSL PREs vs residue number. Upper bar is colored by the PBP5 domain; residues with reduced PRE values are colored cyan. **G** PBP5 S459C PRE residues with reduced PREs are plotted on the structure PBP5 (PDBid: 6MKA) as cyan balls. Static distances between domains indicated. **H** The structures of the open and closed states of PBP5 (open state: N1, light blue, N2, dark gray, nPB, light gray and TP, light green; closed state: N1, light cyan; N2, gray; nPB and TP overlap identically with the open state). PBP5 domain dynamics were observed (green arrows) and not observed (red arrows with an 'X') in solution using PRE experiments. Source data are provided as a Source Data file.

T620 (Fig. 1B, *right*). The active site groove is enclosed above by the 'lid' (aa 443–471) and below by the C-terminal helix (aa 655–678), which together define the deep β-lactam binding cleft (Fig. 1B, *left*; Supplementary Fig. 1C–E)[14].

In class B PBP transpeptidases, the catalytic serine attacks the carbonyl of the penultimate ᴅ-Ala residue of an acyl donor stem peptide, releasing the C-terminal ᴅ-Ala and forming a covalent acyl-enzyme adduct with the donor peptide[15]. Based on their structural resemblance, β-lactams are analogs of the ᴅ-Ala-ᴅ-Ala extremity of donor stem peptides and act as suicide substrates by acylating the Ser nucleophile of PBPs[16]. The stability of the resulting acyl-enzyme complex accounts for the irreversible inactivation of the PBPs in the time scale of a bacterial generation. In the second step, the carbonyl of the

ᴅ-Ala adduct undergoes nucleophilic attack from a primary amine located at the extremity of the side chain of an acceptor stem peptide[13]. This creates a bridge between the peptides and, in turn, links the gly-can strands to one another[17]. Since their discovery as the targets of β-lactam antibiotics, PBPs have been the subject of intense research, especially regarding their role in the resistance to β-lactams of both *S. aureus* and *enterococci*[10–12]. Over the last two decades, a variety of PBP structures have been reported[18–21]. Furthermore, many mutations involved in PBP-mediated β-lactam resistance have been characterized, but no significant structural differences have been identified that allow for a comprehensive understanding of both reduced interaction with β-lactams and effective transpeptidation[22,23]. Studies of PBP2x variants from *Streptococcus pneumoniae* suggested that a lack of a water

molecule coordinated near the catalytic site may facilitate the hydrolysis of bound antibiotics; however, no detailed mode of action was defined[24].

Here we use biochemistry, crystallography, and solution-state biomolecular NMR spectroscopy to define the molecular basis of β-lactam resistance mediated by *E. faecium* PBP5. Due to its large, ~70 kDa size, PBP5 is a challenging target for NMR spectroscopy. Thus, we established a robust protocol for the expression and purification of PBP5 that resulted in exceptional spectra quality, allowing these studies to be achieved. After demonstrating that the N1 and N2 domains of PBP5 are highly mobile in solution, we showed that PBP5 contains only a single binding site for β-lactams, i.e., the active site. Using a penicillin-binding assay, we quantified the extent of resistance in clinically identified PBP5 resistance variants and determined the 3D structures of more than 8 variants, including the most and least resistant, with and without penicillin. As has been observed for other PBPs and their variants, the structures were nearly identical, suggesting that the amino acid substitutions responsible for PBP5-mediated β-lactam resistance impact biophysical properties of PBP5 beyond structure, possibly protein dynamics. Thus, we tested PBP5 dynamics in solution by recording $^{13}$C methyl relaxation data on isoleucine, leucine, and valine (ILV) residues in PBP5, as well as several of its resistant variants. Systematic and comprehensive analysis showed that the most resistant variants have increased dynamics in either their free or penicillin-bound forms. Most interestingly, the PBP5 variants with the most increased dynamics upon penicillin-binding also had the highest penicillin hydrolysis rates, highlighting that it is likely that the increased dynamics lead to increased hydrolysis of the penicillin–PBP5 adduct.

## Results

### Analysis of *Enterococcus faecium* PBP5 by NMR spectroscopy

To understand how mutations in PBP5 confer resistance to β-lactam antibiotics, we used solution-state biomolecular NMR spectroscopy. PBP5 (678 aa; 73.6 kDa) lacking the N-terminal transmembrane domain (residues 1-36), was used for all structural and functional studies (Fig. 1A, B, Supplementary Fig. 2A). PBP5$_{37-678}$ (soluble PBP5; 641 aa; 69.7 kDa; $T_m = 333$ K; for all NMR assignment studies PBP5 T485 was used) is a large protein for biomolecular NMR spectroscopy. Thus, PBP5$_{T485}$ expression and purification were optimized to maximize yield (~40 mg/L of $^2$H-labeled PBP5$_{T485}$), after which NMR buffer conditions were optimized to maximize spectral quality and protein stability (differential scanning fluorimetry [DSF]; 10 mM MES pH 5.8, 25 mM NaCl; low salt ensures the highest sensitivity NMR data measured using cryogenically cooled NMR probes). After ~40 days at 25 °C, ~90 NH cross-peaks were still present in a H/D exchange experiment, demonstrating that portions of the hydrophobic core of PBP5$_{T485}$ are inaccessible to solvent (Supplementary Fig. 2B). To facilitate amide H/D exchange in these regions, we established a protocol for denaturing and refolding $^2$H-labeled PBP5$_{T485}$ in $^1$H-containing buffers (the crystal structure of refolded PBP5$_{T485}$ is identical to PBP5$_{T485}$ purified without refolding, root mean square deviation [RMSD] 0.31 Å$^2$, Supplementary Table 1, Supplementary Fig. 2C). Together, these optimization efforts enabled us to record 2D [$^1$H,$^{15}$N] transverse relaxation optimized spectroscopy (TROSY) data of PBP5$_{T485}$ of remarkable quality (Fig. 1C), allowing the sequence-specific backbone and the side chain methyl group (Isoleucine, Leucine, and Valine [ILV]) assignments of PBP5 to be determined (the latter assignment was confirmed by >70 single point variants; Supplementary Table 2, Supplementary Figs. 2A, 3A). The PBP5$_{T485}$ chemical shift index (CSI) showed that the secondary structure of PBP5$_{T485}$ in solution agrees well with that observed in the PBP5$_{T485}$ crystal structure (Supplementary Fig. 3B, C).

### Interdomain flexibility of PBP5

Crystal structures of PBP5$_{T485}$ and other low-affinity class B PBPs have shown that N1 and N2 domains can rotate as rigid bodies independently

of the nPB and PB domains[14]. To understand the extent of these inter-domain dynamics in solution, we used paramagnetic relaxation enhancement (PRE) measurements[25]. Briefly, site-specific attachment of a paramagnetic spin label (MTSL [S-(1-oxyl-2,2,5,5-tetramethyl-2,5-dihydro-1H-pyrrol-3-yl)methyl methanesulfonothioate] covalently attached to a single cysteine residue) enhances the transverse relaxation rates (proton–electron dipole–dipole relaxation) of nuclei within ~15–20 Å of the label, leading to line broadening. Because PBP5 lacks cysteines, PBP5 variants suitable for MTSL labeling required only a single mutation, i.e., the introduction of the Cys residue. Three variants were generated, one for the N2 domain (S231C) and two for the TP domain (S459C and Q520C). S231 is located on the bottom of the N2 domain, while S459 is located at the top of the TP domain, in the 'lid' that defines the top of the PBP5 active site cleft, and Q520 is located in the middle of the TP domain, on one of the two extended loops, L1 and L2, that bridge the TP and nPB domains.

Both S231C and S459C showed only small, local differences in chemical shifts (chemical shift perturbations, CSPs) compared to PBP5$_{T485}$, demonstrating that the mutations minimally impacted the overall PBP5 structure, and thus we proceeded with MTSL labeling (Supplementary Fig. 4A–D). PRE measurements with PBP5 S231C-MSTL showed that the N2 domain is mobile, as the N2-bound MTSL resulted in line broadening of cross peaks from residues in both the N1 and nPB domains, some of which are nearly 60 Å away from S231C in the open state structure (Fig. 1D, E). This demonstrates that, in solution, the PBP5 N1 and N2 domains are mobile, adopting a range of orientations that allows for approaches close enough for paramagnetic relaxation enhancement. However, this mobility is still constrained, as none of the residues in the TP domain were impacted by S231C MTSL-labeling. Thus, while the N2 domain is clearly mobile, its mobility is constrained to the region below the nPB domain, i.e., it is not able to rotate 'upward' closer to the TP domain. Consistent with this result, PRE measurements with S459C MTSL-labeled PBP5 showed that only residues immediately surrounding S459C, all within the TP domain, were influenced (Fig. 1F, G). No impact on N1 or N2 residues was observed. Together, these data show that in solution PBP1 domains N1 and N2 are mobile, but their mobility is restricted to the region below the nPB domain (Fig. 1H).

In contrast to S231C and S459C, the mutation of Q520 to a cysteine (Q520C) had a significant impact on the 2D [$^1$H,$^{15}$N] TROSY spectrum (Supplementary Fig. 4E–G), resulting in CSPs throughout the spectrum. Although we were unable to leverage this PBP5 variant for MTSL-labeling due to the multiple CSPs, this observation suggested that elements of the intricate fold of PBP5 mediate communication between the TP and N1 domains, possibly via the extended loops L1 and L2 (Supplementary Fig. 4H). To test this, we investigated chemical shift changes in PBP5 variants that had previously been used to confirm the ILV assignment (Supplementary Fig. 2A). While mutations in the TP (I483A, V468A) and N2 (V239A) domain resulted in localized CSPs (Supplementary Fig. 5), mutations in the L1 (I514A, I521A) and L2 (I569A, I581A) loops, like that of Q520C, resulted in CSPs in cross peaks corresponding to residues from both the TP and N1 domains (Supplementary Fig. 6). Together, these data suggest that elements of the interconnected fold of PBP5 especially loops L1 and L2, facilitate communication between the TP and N1 domains.

### The interaction of PBP5 with β-lactam antibiotics in solution

β-lactams mimic the D-Ala–D-Ala extremity of the physiological substrate, i.e., the peptidoglycan precursor, resulting in the covalent acylation of the PBP5 nucleophile S422. To characterize how PBP5 interacts with β-lactams in solution, we plotted the CSPs (detected in 2D [$^1$H,$^{15}$N] TROSY spectra) between free PBP5 and PBP5 titrated with saturating concentrations of either penicillin G (benzylpenicillin, penG) or the fifth-generation anti-MRSA cephalosporin (ceftaroline, Supplementary Figs. 1D, E, 7). CSPs report on changes in the local

environment upon ligand engagement and result from either a direct interaction (binding directly to ligand) or an indirect interaction (conformational or dynamic changes in PBP5 due to β-lactam binding). As expected, the most prominent changes occur in the TP domain due to direct interactions of penG and ceftaroline at the PBP5 β-lactam binding groove (Supplementary Figs. 7, 8A). However, instead of fully lining the PBP5 β-lactam binding groove, as might have been expected based on the structures of multiple PBP5 acyl-enzyme complexes with β-lactams (Supplementary Fig. 1D), the majority of CSPs occur in residues that are above and behind the β-lactam-binding site, indicating that the β-lactams, upon acylation of Ser422, affect these areas (Supplementary Fig. 8A). Interestingly, perturbed residues include positions that vary in clinically resistant isolates of *E. faecium* associated with mutations conferring resistance (i.e., Y460A, T499I)[26].

For both β-lactams, a small number of CSPs were also observed in the nPB and N1 domains. CSPs occur as a result of either direct (second β-lactam-binding site) or indirect effects (i.e., conformational changes in PBP5 upon β-lactam binding and/or changes in PBP5 dynamics). The previously determined crystal structures of free PBP5 and the PBP5-penG acylated adduct are essentially identical (RMSD 0.7 Å[2], TP domain)[14]; thus, the observed differences are not due to a change in PBP5 conformation. Similarly, while PBP5$_{T485}$ robustly binds a fluorescent derivative of penicillin (BOCILLIN FL), no binding is observed with the catalytically inactive S422A variant (PBP5$_{S422A}$ structure is identical to PBP5, Supplementary Table 1; Supplementary Fig. 8B). These data demonstrate that there is only a single binding site for β-lactams in PBP5, i.e., the catalytic nucleophile S422. To further exclude the presence of a second binding site, we mutated three residues at the N1-nPB interface, a site previously suggested to weakly bind β-lactams and substrates in *S. aureus* PBP2a, by altering/swapping their charge: D285K, N289K and E291K (Supplementary Figs. 2A, 8C). In contrast to the S422A variant, the D285K/N289K/E291K variants bind BOCILLIN FL as effectively and with the same kinetics as PBP5$_{T485}$ (Supplementary Fig. 8D). Finally, as a third test, we purified two PBP5 variants that exhibit significant CSPs in N1 (L56I, I83V; mutations to Ala do not fold). These variants also showed no difference in BOCILLIN FL binding compared to PBP5$_{T485}$ (Supplementary Fig. 8E). Together, these data show that β-lactams bind exclusively to the TP domain at the active site and, furthermore, that mutating residues in the N1 or nPB domains do not alter the extent or kinetics of BOCILLIN FL active site binding.

## Clinically derived resistant PBP5 variants do not impact the PBP5 structure

Previous studies established that higher levels of resistance in *E. faecium* is due, in part, to the interaction of β-lactams with PBP5 variants[26–33]. To dissect the molecular basis of this resistance, we generated 17 PBP5 variants that were either identified in nosocomial resistant *E. faecium* isolates[26,33–35] or hypothesized to be β-lactam sensitive (e.g. additional insertion variants in the lid surrounding iS466 [i indicates a serine insertion after residue S466]) and used a BOCILLIN FL binding assay to quantify the extent of acylation in comparison to that of PBP5$_{M485}$, which is the least resistant PBP5 variant reported[27]. The data show that resistant PBP5 variants exhibit an up to ~8-fold less acylation compared to PBP5$_{M485}$ (Fig. 2A). Furthermore, they also show that the PBP5 residue with the most significant impact on BOCILLIN FL acylation is 485. Specifically, PBP5$_{A485}$ binds BOCILLIN FL much weaker than PBP5$_{M485}$. Critically, these results correlate with previously published β-lactam MIC data for mutagenized PBP5 genes expressed in *E. faecium*, with the lowest MIC level values observed for variants with PBP5$_{M485}$ and the highest MIC values observed for variants with PBP5$_{A485}$[26]. These data demonstrate that the mutations in PBP5 contribute to resistance in the *E. faecium* strains in which they are present. To test if the insertion location was important for the observed resistance of the iS466 insertion, we also generated the non-clinical insertion variants iS467, iS468, iS469, iS470, and iS471. Compared to iS466, none of these non-clinical insertion variants exhibited substantial resistance, highlighting the importance of the insertion position (i.e., after residue 466) for negatively impacting the variant's susceptibility to β-lactam antibiotics.

To determine if the observed differential extent of acylation is due to a change in conformation, we used X-ray crystallography (Supplementary Table 1). First, we determined the structures of eight of the most or least resistant variants (iS466/A485, A485, R464A, iS466, Y460A, M485/T499I/V629E, M485M/T499I, M485) and compared the conformations of their individual domains. The structures of the variants are essentially identical (Fig. 2B). This is also true for the highly resistant iS466 variants, in which the additional serine does not result in large conformational changes in the TP domain, but rather creates a short kink into a surface loop (Fig. 2C). The largest differences in the structure were confined to the two loops previously identified to adopt either a range of conformations (Loop629, 'L629', PBP5 residues 622–633) or undergo small rigid body rotations via a hinge motion (the 'lid', residues 447–470). We also determined the structures of the acyl-

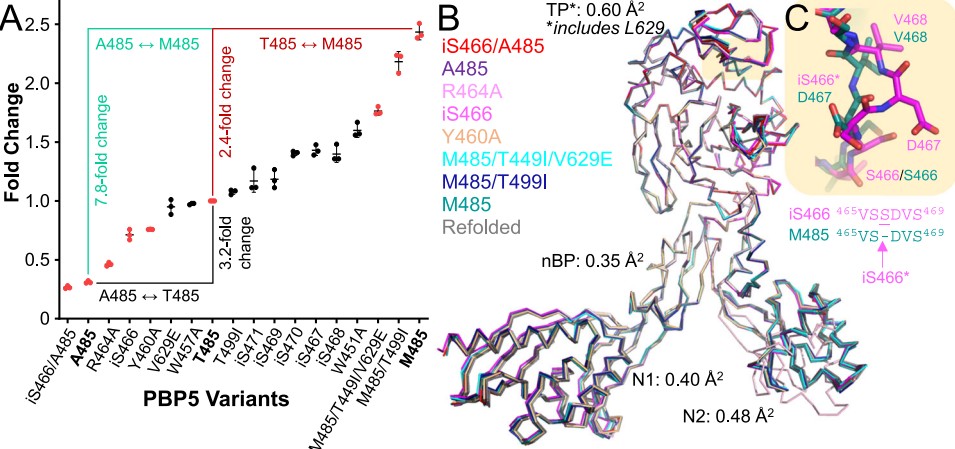

**Fig. 2 | PBP5 clinical resistance variants. A** BOCILLIN FL binding analysis (SDS–PAGE) of PBP5 clinical variants. *n* = 3; average ± std deviation. PBP5 variants with determined 3D structures are highlighted in red. PBP5 A485/T485/M485 bolded. **B** Overlay of 9 PBP5 variant structures with the domain RMSDs reported; no large conformational changes can be detected. **C** Close-up of the S466 loop, illustrating the 'kink' caused by the insertion of an extra serine (iS466*) after residue S466. S466 loop residues from representative PBP5 variants with (iS466, magenta) and without (M485, teal) are shown. Source data are provided as a Source Data file.

enzymes of seven of the same variants bound to penG (Supplementary Table 1). Again, no major differences, except for the L629 loop and the lid, were observed. These data show that the clinical variants do not significantly alter the conformation of PBP5, suggesting that the observed differential affinities of the PBP5 variants for BOCILLIN FL may be a consequence of altered PBP5 dynamics.

## PBP5 dynamics

To further define the dynamics of PBP5$_{T485}$, especially loops like L629 and the lid, we used auto-correlated $^{15}$N fast timescale NMR relaxation measurements (Supplementary Fig. 9A, B). These dynamics data show that PBP5 is a largely rigid protein, with significantly increased ps/ns dynamics only in the N2 domain and short loops that include L629, but not the lid. The increased dynamics in the N2 domain confirm the extensive mobility of this domain in solution, a result not only consistent with our PRE data (Fig. 1E) but also the crystallographic data, as the electron density for this domain is typically the least well-resolved. We also generated the TP deletion construct, PBP5 37–409, which lacks the TP domain, and recorded a 2D [$^1$H,$^{15}$N] TROSY spectrum (Supplementary Fig. 9C, D). Overlaying this spectrum with the PBP5 2D [$^1$H,$^{15}$N] TROSY spectrum shows that while the N2 domain is folded, the rest of PBP5 fails to fold (Supplementary Fig. 9D). These data show that the N1, nPB, and TP domains function as a single folding unit, while the N2 domain folds independently. To determine if β-lactam acylation affects PBP5 loop dynamics, we repeated the auto-correlated $^{15}$N fast timescale NMR relaxation measurements of PBP5 in the presence of penG (1:8 ratio; saturating conditions). No changes in the ps/ns dynamics of the lid or L629 were detectable, demonstrating that penG binding does not alter the dynamics of these loops (Supplementary Fig. 9E, F).

## Resistant variants exhibit increased dynamics in free or penG-bound PBP5

To test if the molecular basis of PBP5 resistance is due to a change in PBP5 dynamics, we used $^{13}$C ILV methyl dynamics experiments, as these NMR measurements are the most rigorous for large proteins such as PBP5. $^{13}$C ILV methyl dynamics are influenced by the type of residue (i.e., Ile vs. Val/Leu) as well as its position within PBP5. Analysis of PBP5$_{T485}$ $^{13}$C ILV methyl dynamics highlighted that PBP5 consists of clusters of rigid and dynamic domains (Supplementary Figs. 10, 11, Supplementary Table 3). Most striking is PBP5 residues 471–556, which form a large rigid cluster in the TP domain and include loops L1 and L2, which connect the PBP5 active site to the nPB and N1 domains (Supplementary Fig. 11).

First, we compared the dynamics of PBP5$_{A485}$ (most resistant), PBP5$_{T485}$, and PBP5$_{M485}$ (least resistant; Supplementary Fig. 12). The data show that more ILV residues show increased flexibility in the PBP5 TP domain as the sidechain of PBP5 residue 485 is lost (M ~ T vs. A; Fig. 3A-E, Supplementary Fig. 13). That is, PBP5$_{A485}$ has increased fast timescale pn/ns dynamics (Fig. 3A, E). We also showed that these increased dynamics are not quenched upon penG binding (Fig. 3B, Supplementary Figs. 12, 13, Supplementary Table 4). Furthermore, using $^{13}$C ILV constant-time Carr−Purcell−Meiboom−Gill (CPMG) measurements that test μs/ms timescale dynamics,[36,37] we showed that PBP5 residues with increased fast timescale dynamics show no change in their μs/ms timescale dynamics (Supplementary Fig. 14). Next, we tested the rate of penG hydrolysis for PBP5$_{485}$ variants by following the change in the penG 1D $^1$H NMR spectra over time. The most resistant and dynamic variant, PBP5$_{A485}$, had the fastest rate of hydrolysis (Fig. 3F, Supplementary Fig. 15A, B). In contrast, the least resistant and least dynamic variant, PBP5$_{M485}$, had the slowest rate of hydrolysis (Fig. 3F, Supplementary Fig. 15B). Together, these data demonstrate that the molecular basis for antibiotic resistance of these PBP5 variants is due to increased protein dynamics that allows for the faster hydrolysis of bound penG.

## The molecular basis of resistance of the clinical insertion variant iS466

To determine if all resistant variants of PBP5 use the same mechanism to achieve resistance, we also tested the clinical PBP5 insertion variant iS466 (an additional serine inserted after S466). Different from PBP5$_{A485}$, PBP5$_{iS466/T485}$ did not have increased dynamics; rather its dynamics were similar to PBP5$_{T485}$ (Fig. 4A, Supplementary Fig. 16). However, upon the addition of penG, the dynamics of PBP5$_{iS466/T485}$ increased significantly (Fig. 4B, Supplementary Fig. 17), similar to those observed for PBP5$_{A485}$ (with or without penG acylation). We then repeated this experiment using PBP5$_{iS466/A485}$. PBP5$_{iS466/A485}$ exhibits dynamics that mirror those of PBP5$_{iS466/T485}$; namely it is only upon the addition of penG that increased dynamics in PBP5$_{iS466/A485}$ are observed (Fig. 4C–G, Supplementary Fig. 16). In a final experiment, we again measured the rate of penG hydrolysis for PBP5$_{iS466}$ and PBP5$_{iS466/A485}$ by following the change in the penG 1D $^1$H NMR spectra over time. Both variants exhibit much-increased hydrolysis when compared to PBP5$_{M485}$ (Fig. 4E, Supplementary Fig. 15C) and even PBP5$_{A485}$. These data highlight the critical contribution of increased protein dynamics detected in these clinical variants that allow for faster hydrolysis of bound penG and subsequently likely account for antibiotic resistance in PBP5.

## Discussion

Increased resistance of ESKAPE pathogens, such as *Enterococcus faecium,* to many common β-lactam antibiotics is a major threat to worldwide health[38,39]. PenG and ceftaroline, two β-lactams that mimic the D-Ala−D-Ala dipeptide substrate moiety[40], bind PBP5 in the deep catalytic cleft via S422 to form a covalent acyl-enzyme complex (Fig. 1B). Our solution data are consistent with this result, as the majority of chemical shift perturbations (CSPs) are localized above and below the β-lactam binding pocket. Furthermore, no second allosteric binding site, as posited to exist in PBP2a from *S. aureus*[41–43], was observed, a result we further confirmed using mutagenesis and BOCILLIN FL binding assays.

The β-lactam binding CSPs occur in regions that are established hotspots for mutations that alter PBP5 β-lactam susceptibility. In particular, our BOCILLIN FL binding assays (Fig. 2A) and previous MIC measurements showed[26] that the clinically relevant PBP5 isolates with mutations in residue 485 and residue iS466 (insertion after S466) are particularly important, as amino acids with shorter side chains at position 485 and insertions after S466 exhibit reduced BOCILLIN FL binding and higher MIC concentrations with β-lactam antibiotics such as ampicillin. The crystal structures of seven distinct PBP5 variants that spanned β-lactam susceptibilities (with and without penG) failed to identify conformational changes that explained the observed differences in β-lactam sensitivity (Fig. 2B). Our crystallography results mirrored those for PBP2 from *N. gonorrhea*[23,44] and PBP2x from *S. pneumoniae*[45], which also did not show major conformational changes to understand their molecular basis of resistance. In these cases, the authors suggested that increased resistance is facilitated by increased flexibility[46,47]. Thus, we leveraged state-of-the-art solution biomolecular NMR spectroscopy, which is the sole technique that can directly measure protein dynamics at multiple timescales.

Our data show that local changes in fast timescale (ps/ns) PBP5 dynamics directly correlate with changes in β-lactam susceptibility. In particular, PBP5 residue 485 variants with shorter sidechains (M → T → A), i.e., those with increased resistance to β-lactams, exhibited increased dynamics (Fig. 3), but showed no changes in intermediate timescale (μs-ms dynamics) as confirmed by $^{13}$C ILV ct-CPMG measurements. We then showed that these resistant and dynamic PBP5 variants exhibit faster penG-acyl-enzyme hydrolysis, a result that fully correlates with the higher MIC of these PBP5 variants. PenG-acyl-enzyme hydrolysis is even faster for the iS466 and iS466/A485 PBP5 variants, where increased dynamics are only seen after the formation

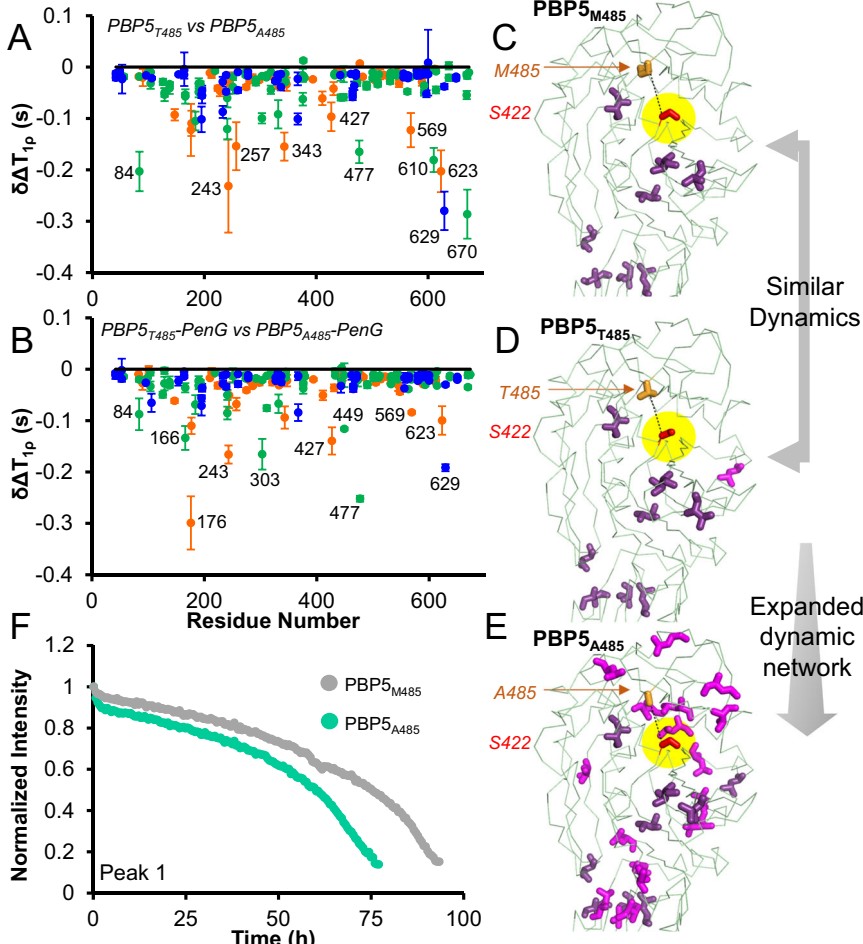

**Fig. 3 | PBP5 A485 has increased dynamics. A** $^{13}$C ILV side chain dynamics ($T_{1\rho}$) comparison between PBP5$_{T485}$ and PBP5$_{A485}$ (delta $T_{1\rho}$ PBP5$_{T485}$–$T_{1\rho}$ PBP5$_{A485}$). Residues with increased dynamics have negative delta $T_{1\rho}$ values and are annotated. Error bars are based on repeat measurements and are often smaller than the symbols used. average ± std deviation. **B** $^{13}$C ILV side chain dynamics ($T_{1\rho}$) comparison between PBP5$_{T485}$:penG (1:8 saturation) and PBP5$_{A485}$:penG (1:8 saturation) (delta $T_{1\rho}$ PBP5$_{T485}$:penG–$T_{1\rho}$ PBP5$_{A485}$:penG). Error bars are based on repeat measurements and are often smaller than the symbols used. average ± std deviation. Changes in $T_{1\rho}$ dynamics mapped on the PBP5 TP domain structure for **C** PBP5$_{M485}$, **D** PBP5$_{T485}$, and **E** PBP5$_{A485}$. The catalytic S422 is shown as sticks, highlighted in yellow and labeled. The 485 residue is shown as orange sticks and labeled. Residues that experience dynamics are shown as sticks and colored in pink (**C**) or pink and magenta, with magenta indicating the residues that differ between M485 and T485 (**D**) or T485 and A485 (**E**). **F** PenG hydrolysis by PBP5$_{A485}$ (green) and PBP5$_{M485}$ (gray) following peak intensity via $^1$H NMR spectroscopy; Peak 1. Source data are provided as a Source Data file.

of the acyl-enzyme complex (Fig. 4), highlighting that PBP5 has developed different molecular routes to gain resistance. Together, these data show that PBP5-resistant variants achieve resistance via an increase in their intrinsic local dynamics which, in turn, enhances the rate of acyl-enzyme hydrolysis. As this step more rapidly regenerates the native, fully active PBP5 enzyme, it is more resistant to β-lactams antibiotics.

Increased acyl-enzyme hydrolysis is likely achieved via an increase in the frequency of access of the water and/or activated and/or oriented toward the acyl bond for productive nucleophilic attack (due, for example, to an increase in water access and/or proper orientation of water-coordinating sidechains). While the structures of the PBP5 variant backbones are essentially identical, a loss of sidechain atoms at position 485 results in the creation of a cavity behind the catalytic Ser422 (Fig. 5A). The increased dynamic network available to variants with this pocket, i.e., PBP5$_{A485}$, likely results in increased access, orientation and activation of the nucleophilic water at the acyl-enzyme bond, contributing to faster hydrolysis (Fig. 5B, *left, middle*). Similarly, while the insertion of an additional serine residue after S466, (iS466) into the lid also results in only a small, local distortion of the PBP5 backbone (Fig. 2C, affecting only residues 466–470), this distortion is located directly above the bound β-lactams (Fig. 5B, *right*, black

dashed arrow). The increased dynamic network available to variants with the iS466 insertion results in even faster β-lactam hydrolysis, potentially by facilitating the orientation of the β-lactam acyl-enzyme for nucleophilic attack (Fig. 5B). Importantly, Ser insertions at PBP5 residues 466–471 do not lead to increased resistance. Collectively, these data show that clinically derived mutations that reduce sidechain bulkiness and/or expand surface loops result in increased dynamics that facilitate acyl-enzyme hydrolysis and, via increased β-lactam turnover, lead to β-lactam resistance. Thus, this study identifies the molecular basis of resistance of multiple clinical *E. faecium* PBP5 variants, results that are likely applicable to the PBP family generally.

## Methods

### Plasmid construction

PBP5$_{37-687}$ (residues 37-678 that lack the N-terminal transmembrane domain) gene from *Enterococcus faecium* was cloned into the pRP1B vector with His$_6$-tag and a Tobacco Etch Virus (TEV) protease cleavage site as previously described[14]. For the assignment of Isoleucine/Leucine/Valine (ILV) residues, the following PBP5 variants were created: V49I, L56I, I83V, L99I, L153I, I176V, L184I, V195A, I213A, V233A, V239A, I243A, L274A, I275A, V278A, I281A, L303A, L332I, I333A, I343A, L345A, I347A, I377A, I427A, I431A, L433A, L438A, I446A, L449A, V462A, V465A,

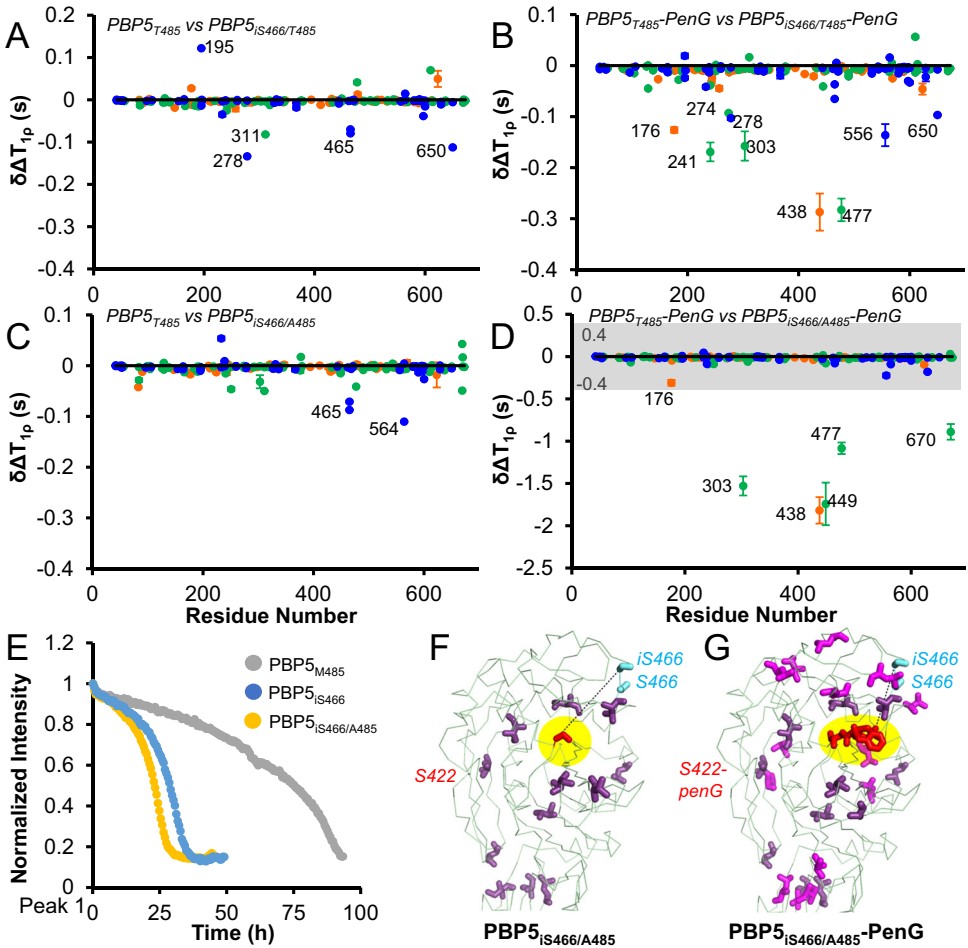

**Fig. 4 | PBP5 iS466 has increased dynamics only as a penG acyl-complex. A** $^{13}C$ ILV side chain dynamics ($T_{1\rho}$) comparison between $PBP5_{T485}$ and $PBP5_{iS466}$ (delta $T_{1\rho}$ $PBP5_{T485}$–$T_{1\rho}$ $PBP5_{iS466}$). Error bars are based on repeat measurements and are often smaller than the symbols used. average ± std deviation. **B** $^{13}C$ ILV side chain dynamics ($T_{1\rho}$) comparison between $PBP5_{T485}$:penG (1:8 saturation) and $PBP5_{iS466}$:penG (1:8 saturation) (delta $T_{1\rho}$ $PBP5_{T485}$:penG–$T_{1\rho}$ $PBP5_{iS466}$:penG) showing that $PBP5_{iS466}$ has increased dynamics when bound to penG. Error bars are based on repeat measurements and are often smaller than the symbols used. average ± std deviation. **C** $^{13}C$ ILV side chain dynamics ($T_{1\rho}$) comparison between $PBP5_{T485}$ and $PBP5_{iS466/A485}$ (delta $T_{1\rho}$ $PBP5_{T485}$–$T_{1\rho}$ $PBP5_{iS466/A485}$). Error bars are based on repeat measurements and are often smaller than the symbols used. average ± std deviation. **D** $^{13}C$ ILV side chain dynamics ($T_{1\rho}$) comparison between $PBP5_{T485}$:penG (1:8 saturation) and $PBP5_{iS466/A485}$:penG (1:8 saturation) (delta $T_{1\rho}$

$PBP5_{T485}$:penG–$T_{1\rho}$ $PBP5_{iS466/A485}$:penG) showing that $PBP5_{iS466/A485}$ has increased dynamics when bound to penG. Gray highlight indicates the scale used in panels (**A–C**). Error bars are based on repeat measurements and are often smaller than the symbols used. average ± std deviation. **E** penG hydrolysis by $PBP5M_{485}$ (gray) $PBP5_{iS466}$ (blue) $PBP5_{iS466/A485}$ (yellow) following peak intensity via $^{1}H$ NMR spectroscopy; Peak 1. Changes in $T_{1\rho}$ dynamics mapped on the PBP5 TP domain structure for **F** $PBP5_{iS466/A485}$ and **G** $PBP5_{iS466/A485}$:penG. The catalytic S422 (F) or S422-PenG (G) are shown as red sticks, highlighted in yellow, and labeled. S466 and inserted serine iS466 are shown as cyan sticks and labeled. Residues that experience dynamics are shown as sticks and colored in pink (**F**) or pink and magenta, with magenta indicating the residues that differ between $PBP5_{iS466/A485}$ and $PBP5_{iS466/A485}$:penG (**G**). Source data are provided as a Source Data file.

V468A, V471A, L477A, I478A, I483A, I505A, L510A, I514A, I521A, I531A, L532A, L543A, I545A, I548A, V556A, L563A, V564A, I569A, V580A, I581A, V586I, I589A, V590L, V596A, V600A, L607A, I612A, L614A, I623A, V629E, L637A, L669A and L670A (Supplementary Table 2). To identify the most resistant PBP5 variants using a BOCILLIN FL assay, the following PBP5 variants were generated (i denotes insertion mutation): iS466/A485, M485A, T485, Y460A, T499I, R464A, W451A, W457A, iS466, iS467, iS468, iS469, iS470, iS471, V629E, M485/T499I, M485/T499I/V629E and M485[26]. For NMR PRE measurements, PBP5 variants S231C, S459C, and Q520C were generated. All the variants were generated using the QuikChange site-directed mutagenesis kit (Agilent) and primers were purchased from Integrated DNA Technologies (IDT).

## Protein expression

PBP5 was expressed and purified as described earlier[14]. Briefly, PBP5 plasmid DNA was transformed into *Escherichia coli* BL21 (DE3) cells

(Agilent). Cells were grown in LB medium in the presence of selective antibiotic (Kanamycin) at 37 °C until an $OD_{600}$ ~ 0.8–1.1, upon when the cells were cooled down to 18 °C prior to the addition of 0.5 mM isopropyl β-D-thiogalactopyranoside (IPTG; GoldBio) for protein induction. Cells were harvested the next day, 18–20 h after induction by centrifugation (8000×$g$ for 15 min). Cell pellets were kept in −80 °C freezer until purification. All PBP5 variants were expressed using the identical process as described above.

The expression of uniformly ($^2H$,$^{15}N$)- or ($^2H$,$^{15}N$,$^{13}C$)-labeled PBP5 or its variants for NMR backbone assignments, titrations, and backbone relaxation measurements was achieved by growing cells in $D_2O$ based M9 minimal media containing 1 g/L $^{15}NH_4Cl$ and/or 4 g/L ($^2H$,$^{13}C$)-D-glucose (CIL or Sigma-Aldrich) as the sole nitrogen and carbon sources, respectively. Multiple rounds (0%, 25%, 50%, 75%, and 100%) of $D_2O$ adaptation were necessary for high-yield expression.

Selective, single amino acid labeling of Phe, Tyr, Leu, and Val residues of PBP5 was accomplished by adding individual

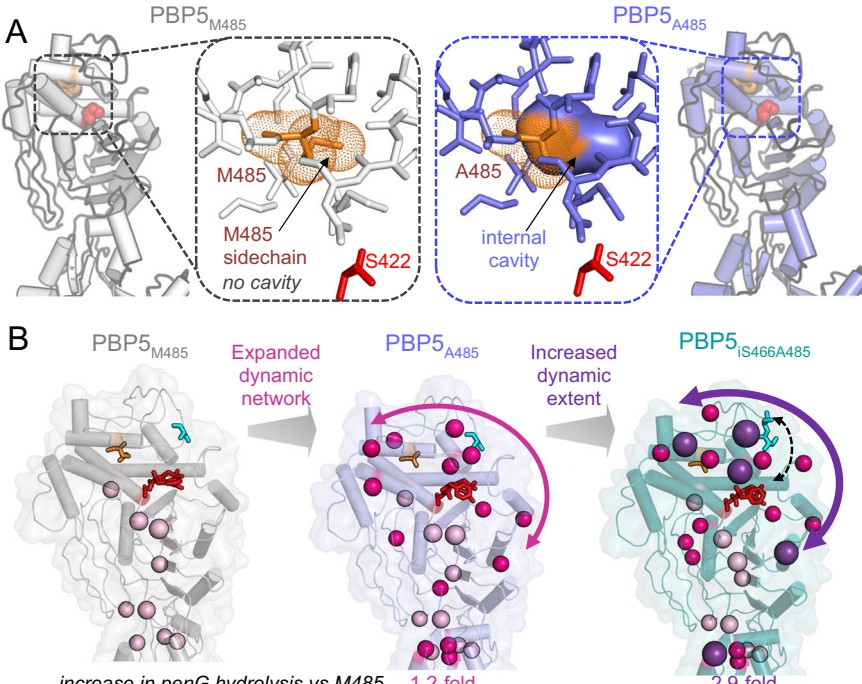

**Fig. 5 | Increased dynamics are correlated with increased β-lactam resistance.** **A** PBP5 residue 485 pockets for PBP5$_{M485}$ (left, gray) and PBP5$_{A485}$ (right, lavender); insets are expanded views of the region boxed. The 485 residue is shown as sticks, with dots defining the van der Waal's radii. The catalytic S422 is shown as red sticks. Internal cavities that are not accessible to solvent are shown as solid surfaces. As can be seen in PBP5$_{A485}$ (right), a large cavity is present due to the short Ala side-chain. In contrast, in PBP5$_{M485}$ (left), the sidechain atoms of the met residue occupy this space. **B** A limited number of residues show dynamics in PBP5$_{M485}$ (left, gray; the Cα atoms of residues with dynamics are highlighted as light pink balls) this network expands significantly (nearly 3-fold) when the residue at 485 is mutated to an alanine (additional peaks shown as dark pink). The addition of an inserted serine residue after S466 (iS466) adds additional dynamics, as well as shows, increased dynamics (purple balls). Both increased dynamics variants have an increase in penG hydrolysis. Source data are provided as a Source Data file.

unlabeled L-amino acids into minimal media in H$_2$O along with $^{15}$N-Phe, $^{15}$N-Tyr, $^{15}$N-Leu, and $^{15}$N-Val amino acids, respectively, to identify the H$^N$/N cross-peaks from these amino acids in the 2D [$^1$H,$^{15}$N] TROSY spectrum of PBP5[48]. For selective single amino acid un-labeling of Lys, Arg, Met, and Asn residues, PBP5 was expressed in D$_2$O-based minimal media with uniformly ($^2$H,$^{15}$N,$^{12}$C)-labeling using $^{15}$N-NH$_4$Cl and $^{12}$C-D-glucose as sole nitrogen and carbon sources, respectively, along with the addition of the necessary unlabeled L-amino acid (Lys, Arg, Met, and Asn) at a concentration of 1 g/L in the media[49].

A uniformly ($^2$H,$^{15}$N)-labeled PBP5 with Ile, Leu, and Val selectively labeled with $^{13}$C was expressed as described above except for the use of $^{12}$C,$^2$H-D-glucose as the sole carbon source and the addition of 50 mg each of uniformly ($^{15}$N,$^{13}$C)-labeled Ile, Leu, and Val (CIL or Sigma) per 1 L of M9 medium 1 h prior to induction[50].

For the preparation of ILV-labeled PBP5 for assignments and various relaxation measurements, multiple labeling schemes were used[51]. For the assignment of PBP5 ILV residues, PBP5 was expressed with 1 g/l $^{15}$NH4Cl, 4 g/l ($^2$H,$^{13}$C)-D-glucose, ($^{13}$C$_5$,3-$^2$H$_1$) 120 mg/l α-ketoisovaleric acid sodium salt (CDLM 4418) and 60 mg/l ($^{13}$C$_4$;3,3-$^2$H$_2$) α-ketobutyric acid sodium salt (CDLM 4611) in 100% D$_2$O. To assist in the assignment, PBP5 variants were expressed in M9 medium and labeled with 120 mg/l ($^{13}$C) α-ketoisovaleric acid (CLM 6821) and 60 mg/l ($^{13}$C$_2$) α-ketobutyric acid (CLM 6820) in 100% D$_2$O. For NOE-based assignment spectra, PBP5 was expressed with 1 g/l $^{15}$NH4Cl, 4 g/liter ($^2$H,$^{12}$C)-D-glucose, 120 mg/l ($^{13}$C$_5$,3-$^2$H$_1$) α-ketoisovaleric acid (CDLM 7317) and 60 mg/l ($^{13}$C$_4$,3,3-$^2$H$_2$) α-ketobutyric acid (CDLM 7318) in 100% D$_2$O. Lastly, for relaxation measurements, PBP5 was expressed with 4 g/l ($^2$H,$^{12}$C)-D-glucose, 120 mg/l (3-$^{13}$C,3-methyl-$^2$H$_2$, 3,4,4,4,-$^2$H$_4$) α-ketoisovaleric acid (CDLM 7354) and 60 mg/l (4-$^{13}$C,4-$^2$H$_2$, 3-$^2$H$_2$) α-ketobutyric acid (CDLM 7353) in 100% D$_2$O.

## Purification

Cell pellets were resuspended in Lysis Buffer (50 mM Tris pH 8.5, 500 mM NaCl, 5 mM imidazole, 0.1% Triton X-100) and lysed by high-pressure homogenization (EmulsiFlex C3; Avestin). The cell lysate was centrifuged at 40,000×*g* for 45 min at 4 °C. The clarified lysate was filtered through 0.22 μm PES filter (Millipore) and loaded onto a pre-equilibrated His-trap gravity column (Cytiva), washed with 100 mL His-tag Buffer A (50 mM Tris pH 8.5, 500 mM NaCl, 5 mM imidazole), before elution with 60% His-tag buffer B (50 mM Tris pH 8.5, 500 mM NaCl, 250 mM imidazole). The protein was dialyzed in the presence of TEV protease (0.3 mg/mL; 1:10 TEV:PBP5) for 48–60 h at 4 °C to cleave the His$_6$-tag in cleavage buffer (25 mM Tris pH 8.0, 250 ml NaCl). A subtraction purification was carried out using a Ni-NTA gravity column to remove the His$_6$-tag and TEV. PBP5 in the flow-through was collected and dialyzed for at least 5 hours in 10 mM Tris pH 8.0, 1.5 M (NH$_4$)$_2$SO$_4$. The dialyzed protein was loaded onto two 5 ml pre-equilibrated HiTrap Phenyl HP hydrophobic interaction columns (Cytiva) and a gradient of decreasing (NH$_4$)$_2$SO$_4$ was used to separate the folded and misfolded PBP5. Properly folded PBP5 (first peak) was collected and immediately dialyzed against NMR buffer (10 mM MES pH 5.8, 25 mM NaCl) or crystallization buffer (10 mM Tris pH 8.5, 800 mM NaCl). Dialyzed PBP5 was concentrated and loaded onto preequilibrated (NMR or crystallization buffer) SEC 200 26/60 column (Cytiva) for the final purification. The fractions corresponding to PBP5 were pooled and concentrated for NMR measurements or storage by snap freezing in liquid nitrogen and kept at −80 °C. All PBP5 variants were purified using the same procedure as described above.

## Protein refolding

Of the 618 expected peaks in the 2D [$^1$H,$^{15}$N] TROSY spectrum of PBP5, 530 are observed. These NH cross peaks are invisible due to a lack of D/H exchange due to the expression of PBP5 in a D$_2$O-based medium.

Time-dependent 2D [¹H,¹⁵N] TROSY experiments of lyophilized ¹⁵N-labeled PBP5 that was re-suspended in D₂O showed that ~90 peaks are still visible after ≥40 days (298 K), explaining the difference in total peak count. Purified PBP5 was diluted to 0.1–0.2 mg/mL in Refolding Buffer (10 mM Tris pH 8.0, 800 mM NaCl) and placed in a SnakeSkin dialysis tubing (3 kDa MWCO; ThermoScientific). The dialysis tube was then placed in Refolding Buffer with 6 M GuHCl overnight for denaturation. The next day, the tube was transferred to Refolding Buffer with 3 M GuHCl. After 24 h, the tube was transferred to Refolding Buffer with 1.5 M GuHCl for 4 h, 0.5 M GuHCl for 4 h, and Refolding Buffer with no GuHCl for 4 h. Finally, the protein was dialyzed into 10 mM Tris pH 8.0, 1.5 M (NH₄)₂SO₄ for HIC and SEC purifications, as described above.

## Crystallization

PBP5 variants in crystallization buffer (10 mM Tris pH 8.5, 800 mM NaCl) were concentrated to 10–17 mg/ml and crystallized in 2.0–3.2 M ammonium sulfate with the pH ranging from 4.0 to 8.0 (hanging drop, vapor diffusion) in Linbro plates (Hampton Research) at room temperature. PenG–acyl-PBP5 complexes were obtained by co-crystallization using the same conditions and incubating the crystals with a 20- to 40-fold molar excess of PenG for 2 h. PBP5 and PenG-PBP5 complexes were cryoprotected (cryoprotectant agents: 4.0 M ammonium sulfate, or paraffin/silicon oil 50:50 or mother liquor supplemented with 10% (v/v) ammonium sulfate and 7.5% (v/v) glycerol), after which they were immediately flash frozen in liquid nitrogen for X-ray diffraction screening.

## Data collection, processing, and solution

X-ray diffraction data for all crystals were collected at SSRL (beamline 12-2). All datasets were processed using AutoXDS[52] and the data was phased using molecular replacement (MR, PHASER as implemented in Phenix[53]) using PDBid 6MKA[14] as a search model. Final models were obtained using iterative rounds of manual building using Coot[54] and refinement (Phenix). Molecular figures were generated using PyMOL[55]. RMSD calculations were performed using LSQKAB in CCP4[56].

## NMR spectroscopy

All NMR data were collected on Bruker Avance Neo 600 or 800 MHz spectrometers equipped with TCI HCN z-gradient cryoprobes at 308 K. NMR measurements of PBP5 (all NMR sequence-specific backbone assignment measurements were performed using PBP5_{T485}) were recorded using (²H,¹⁵N)-labeled PBP5 at a final concentration of 0.1–0.95 mM in NMR buffer (10 mM MES pH 5.8, 25 mM NaCl) and 90% H₂O/10% D₂O. Data were processed using Topspin 4.1.3 (Bruker) and analyzed using CcpNMR[57], NMRFAM-SPARKY[58], and CARA. The sequence-specific backbone assignments of (²H,¹³C,¹⁵N)-labeled PBP5 (0.95 mM) in the presence and absence of penG were achieved by using 3D triple resonance experiments including 2D [¹H,¹⁵N] TROSY, 3D TROSY-HNCA, 3D TROSY-HN(CO)CA, 3D TROSY-HNCO, 3D TROSY-HN(CA)CO, 3D TROSY-HN(CO)CACB and 3D TROSY-HNCACB. Furthermore, 3D TROSY-HNCA, 3D TROSY-HNCACB along with the 2D ¹H-¹⁵N planes of 3D-HNCA and 3D-HNCO were collected on the uniformly [²H,¹⁵N]-labeled PBP5 with Ile, Leu, and Val selectively labeled with ¹³C. A 2D [¹H,¹⁵N] TROSY was collected on selectively single amino acid labeled and selectively unlabeled samples of PBP5. For PRE measurements, two 2D [¹H,¹⁵N] TROSY spectra were acquired on the PBP5 variants S231C and S459C in the presence and absence of the spin label (MTSL; S-(1-oxyl-2,2,5,5,-tetramethyl-2,5-dihydro-1H-pyrrol-3-yl)methyl methanesulfonothioate; Sigma) to facilitate the assignment. Peak intensities were measured and plotted as the ratio of intensities in the presence of MTSL (active and quenched with ascorbic acid) to confirm the residues in the vicinity of the MTSL probe.

The ILV methyl group assignments were performed using 3D (¹H-¹H) NOESY-(¹H-¹⁵N) HSQC (τ_m = 150 ms), 3D (H)CCH-TOCSY (τ_m = 12 ms), 3D HSQC-NOESY-HMQC (τ_m = 150 ms) and 2D [¹H,¹³C] HMQC experiments from all point mutations PBP5 variants (Supplementary Table 2).

## NMR analysis of Penicillin G and ceftaroline inhibitor binding

PenG (Goldbio) and ceftaroline were titrated into 0.5 mM (²H,¹⁵N)-labeled and 0.1 mM (²H,¹⁵N) ILV labeled PBP5-WT protein at molar ratios of 0:1, 1:1, 2:1, 4:1, and 8:1 (penG/ceftaroline:PBP5). 2D [¹H,¹⁵N] TROSY or 2D [¹H,¹³C] HMQC spectra were recorded for each titration point. PenG was solubilized in 10 mM MES pH 5.8, 25 mM NaCl and ceftaroline was solubilized in d₆-DMSO (200 mM stock). No significant chemical shift differences were identified in the PBP5 2D [¹H,¹⁵N] TROSY or 2D [¹H,¹³C] HMQC spectrum upon the addition of d₆-DMSO. Chemical shift perturbations (Δδ) between apo PBP5 and inhibitor-bound PBP5 spectra were calculated using the below equations

$$\Delta\delta(ppm) = \sqrt{\left(\Delta\delta_H\right)^2 + \left(\frac{\Delta\delta_X}{Y}\right)^2} \qquad (1)$$

where $X$ is ¹⁵N or ¹³C and $Y$ is the constant i.e., 10 for ¹⁵N and 4 for ¹³C.

Residues with significant CSPs were identified by calculating the standard deviation of all CSPs ($\sigma$), then excluding any CSPs larger than average + 3σ, recalculating σ as a new corrected $\sigma_0$, and then repetitively removing CSPs above $3\sigma_0$ until no additional CSPs can be removed. CSPs above $\geq 3\sigma_4$ were then selected as the threshold for ¹H/¹⁵N backbone CSP analysis and $\geq 3 \times \sigma_5$ for ¹H/¹³C methyl ILV data[59].

## ¹⁵N relaxation measurements and analysis

Relaxation measurements were performed on (²H,¹⁵N)-labeled ligand-free and penG bound PBP5 (at saturating 1:8 ratios of PBP5:penG) at a final concentration of 0.5 mM in NMR buffer and 90% H₂O/10% D₂O at 308 K. ¹⁵N longitudinal ($R_1$) and transverse ($R_2$) relaxation rates measurements were acquired using sensitivity-enhanced experiments. TROSY versions of $R_1$ and $R_2$ experiments were acquired on a Bruker Neo 600 MHz spectrometer with a TCI HCN Z-gradient cryoprobe at 308 K using a recycle delay of 4 s between experiments and the following relaxation delays for $T_1$: 100, 500, 1000, 2000, 2300, 2600, 3000, 4000, and 5000 ms (1000 and 2600 ms were repeated for measurement error assessment); and $T_2$: 4.24, 8.48, 12.72, 16.96, 21.2, 25.44, 33.92, 42.4, and 50.88 ms (16.96 and 33.92 ms were repeated for measurement error assessment). 2D [¹H,¹⁵N] TROSY spectra before and after the relaxation measurements ensured that PBP5 was always in saturation with penG and thus no ligand exchange was influencing the data. $T_1$, $T_2$ values were calculated using NMRpipe by fitting the intensity of peaks to the exponential decay function, and errors were determined via relaxation curve fitting.

## ¹³C-methyl relaxation measurements

$T_1$ and $T_{1\rho}$ experiments[60,61] were recorded on uniformly (²H,¹²C,¹⁵N)-labeled PBP5_{T485} and other variants with ¹³CHD₂ labeled ILV methyl groups, either free or with penG or ceftaroline saturated (1:8 ratio), at a final protein concentration of 0.5 mM in NMR buffer and 100% D₂O. Sample concentration was tightly monitored to ensure no effect on $\tau_c$ and thus $T_{1\rho}$ measurements. PenG or ceftaroline inhibitor was titrated to achieve full saturation and saturation was checked between and after relaxation measurements. All relaxation data was recorded as a pseudo-3D in a fully interleaved manner at 308 K. $T_1$ relaxation delays: 20, 500, 1000, 1200, 1500, 1800, 2000, 2500, 3000, and 4000 ms (D1 [recycle delay] of 4 s; 800 MHz; 1200 and 2500 ms were repeated for measurement error assessment). $T_{1\rho}$ relaxation delays: 5, 30, 40, 60, 80, 100, 120, 140, 160, and 180 (D1 [recycle delay] of 3 s; 800 MHz; 40 ms and 140 ms was repeated for measurement error assessment).

Constant-time Carr–Purcell–Meiboom–Gill (ct-CPMG)[61] relaxation dispersion experiments were performed at 308 K at 18.8 T magnetic field strength for PBP5_{T485}, PBP5_{T485}:PenG, PBP5_{A485}:PenG, PBP5_{iS466/A485} and PBP5_{iS466/A485}:PenG. A constant time of 40 ms between ¹³C

refocusing pulses and 10 different delay times corresponding to the following CPMG frequencies 0, 50, 100, 250, 400, 600, 800, 1200, 1600, and 2000 Hz were used (0, 100, and 1200 Hz were repeated for measurement error assessment). D1 [recycle delay] was set to 4.5 s. Upon saturation (chemical shifts of interacting residues stopped changing position in the spectrum; -1:1 ratio), additional PenG (to 1:8 ratio) was added to ensure that all experiments were performed under fully inhibitor-saturated conditions and thus that the observed ct-CPMG dispersions are independent of ligand on/off-exchange events.

### $^{13}C$-methyl (CHD$_2$) relaxation analysis

$T_1$ and $T_{1\rho}$ values were calculated using NMRviewJ[62] using the peak intensities (jitter function) and exponential decay fitting function. Errors were also determined via relaxation curve fitting. $T_2$ was extracted from $T_{1\rho}$ by ($R_{1/2} = 1/T_{1/2}$): $R_2 = (R_{1\rho} - R_1\cos^2\beta)/\sin^2\beta$, where $\beta$ is the effective rotation angle for each $^{13}C$ nucleus as determined by the strength of the spin-lock field and the chemical shift offset of the nucleus from the spin-lock frequency. ct-CPMG relaxation dispersion intensity measurements were performed in NMRviewJ (jitter function) and converted to $R_{2\text{eff}}$ by: $R_{2\text{eff}}(\vartheta_{\text{CPMG}}) = (-1/T_{\text{relax}})\ln(I_{\text{CPMG}}/I_0)$.

### SDS-PAGE PBP5 binding assay

Purified PBP5 proteins and BOCILLIN FL (ThermoFisher Scientific) were diluted to 100 μg/mL in assay buffer (10 mM Tris pH 8.5, 800 mM NaCl). PBP5 proteins (5 μg) were incubated with BOCILLIN FL (5 μg) for 30 min at 37 °C. After incubation, 2x SDS-loading dye was added to the samples followed by denaturation (10 min, 95 °C). Samples were analyzed by SDS-PAGE. BOCILLIN FL-acylated PBP5 protein was visualized using a ChemiDoc MP (BioRad) and the StarBright520 application. SDS-PAGE gels were stained with Coomassie to visualize protein load. Densitometric analysis was performed using Image Lab (BioRad). Three replicates for each BOCILLIN FL-labeled protein sample were used. The average volume of each selected band was measured and normalized to the average volume of protein bands measured after Coomassie staining to account for possible small differences in protein loading. The fold-change in BOCILLIN FL-binding was computed as $(A/B) - 1$, where $A$ is the normalized average volume of the PBP5 variants and $B$ is the normalized average volume of PBP5$_{M485}$.

### BOCILLIN FL fluorescence polarization assay

BOCILLIN FL (Thermo Fisher Scientific, B13233) was dissolved in 100% DMSO (1 mM stock solution) and stored at −20 °C. Assay was performed in a low-volume 384-well black polystyrene assay plate (Corning, 4511) in assay buffer (100 mM sodium phosphate pH 7.0, 0.01% Triton X-100). Fresh BOCILLIN FL working stock was prepared for each assay. BOCILLIN FL (30 nM) and PBP5 (3.6 μM) were used for this assay. To ensure identical concentrations of wt PBP5 and variants used, the concentrations of PBP5 were quantified using the Pierce 660 (Thermo Fisher Scientific) protein assay. Reactions were initiated with the addition of BOCILLIN FL to the plate and the plate was immediately sealed using ThermalSeal RTS silicone adhesive film (EXCEL Scientific, TSS-RTQ-100) to prevent evaporation during the duration of the measurements. Reactions without PBP5 (BOCILLIN FL only) were used as a negative control. The assay was carried out at 37 °C using a CLARIOStar plate reader (BMG Labtech) equipped with a 482 nm excitation filter, a 530 nm emission filter, and a dichroic LP504 nm filter. The focal height was set to 11.7 mm and the gain was set to 10% using the negative control. Measurements were made at 40 s intervals for up to 60 min with 100 flashes per measurement. Three replicates were made for each PBP5 variant in each run and the experiments were repeated 3 times for statistical analysis.

### NMR-based penG hydrolysis analysis

NMR based penG hydrolysis was performed in the presence of PBP5$_{T485}$, PBP5$_{M485}$, PBP5$_{iS466}$, PBP5$_{iS466/A485}$, and PBP5$_{A485}$ PBP5

(50 μM protein concentration) in NMR hydrolysis buffer (20 mM Na-phosphate pH 6.3, 150 mM NaCl, 5% D$_2$O). 500 μM penG (1:10 protein:penG ratio) was used to initiate the reaction with a fully occupied covalent acyl-enzyme adduct complex. Hydrolysis of the acyl-enzyme (up to 4 days) was monitored every 30 min by 1D $^1H$ NMR spectroscopy (water suppression via excitation sculpting with gradients) using a Bruker Neo 600 MHz spectrometer with a TCI z-gradient cryoprobe at 308 K. The high protein:penG ratio allowed for a better equilibration and visualization of the fast hydrolysis and explains the biphasic look of the penG hydrolysis curve. Comparison was achieved by simple end-point analysis.

### Reporting summary

Further information on research design is available in the Nature Portfolio Reporting Summary linked to this article.

### Data availability

All NMR chemical shifts have been deposited in the BioMagResBank BMRB 51690 (sequence-specific $^1H$, $^{13}C$, and $^{15}N$ backbone resonance assignments of 70 kDa Penicillin Binding Protein PBP5) and BMRB 51692; (Sidechain Ile, Leu, and Val methyl chemical shift assignments for Penicillin Binding Protein 5). Atomic coordinates and structure factors have been deposited in the Protein Data Bank (PDB: 8F3F, 8F3G, 8F3H, 8F3I, 8F3J, 8F3L, 8F3M, 8F3N, 8F3O, 8F3P, 8F3Q, 8F3R, 8F3S, 8F3T, 8F3U, 8F3Z, 8F67, 6MKA, 6MKG). Supplementary Figure 18 shows 2Fo-Fc maps for all experimentally determined structures. All other data generated by this study are provided as source data with this paper and are also available at https://doi.org/10.6084/m9.figshare.6025748.v1. Source data are provided with this paper.

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

## Acknowledgements

We thank Mr. J.A. Bray and Dr. Gautam Srivastava for their help with protein expression and purification. We thank Dr. Michael Clarkson for helping with the NMR data collection. The authors would like to thank staff members at the Stanford Synchrotron Radiation Light source (SSRL), SLAC National Accelerator Laboratory for access to X-ray beamlines. Use of the Stanford Synchrotron Radiation Lightsource, SLAC National Accelerator Laboratory, is supported by the U.S. Department of Energy, Office of Science, Office of Basic Energy Sciences under Contract No. DE-AC02-76SF00515. This work is supported by grant 1R01AI141522 from the National Institute of Allergy and Infectious Diseases to W.P. and L.B.R.

## Author contributions

R.P., L.B.R., and W.P. developed the concept. Y.H. performed ILV assignment and NMR data analysis of all 13C ILV relaxation data. Y.H., G.S.K., M.S.C., A.S.S., E.D.D., M.V.S., C.D. cloned, expressed, and purified PBP5 (for all different experiments and labeling schemes). EDD, MVS, and MSC determined crystal structures. G.S.K., M.V.S., and E.D.D. performed and evaluated MTSL experiments. GSK performed sequence-specific backbone assignment and analyzed 15N relaxation data. MVS performed SDS–PAGE assay and MSC performed fluorescence binding assays. W.P., R.P., Y.H., L.B.R., and M.A. wrote the manuscript with comments and inputs from all co-authors.

## Competing interests

The authors declare no competing interests.
