## [Peer Review File · Nature Communications]

REVIEWER COMMENTS

Reviewer #1 (Remarks to the Author):

I find the work by Hunashal et al. truly impressive as far as the technicalities of NMR spectroscopy are concerned. Although I cannot judge the importance or relevance of this work from biological or medical perspective, I can attest with full confidence that the presented NMR data is of superb quality considering the size (molecular weight) of the system (70 kDa). The work seems to be performed with utmost care and is well documented and illustrated. Even though no attempt was made by the authors to extract the actual parameters of dynamics (such as order parameters of N-H or C-H bond vectors or correlation times of various motions), as the material presented is of comparative nature (i.e. only the changes in relaxation parameters are tracked), I believe that the authors' conclusions are sound and would enthusiastically recommend publication of this work provided that it satisfies the requirements of Nature Communications with respect to biological/medical novelty.

The following minor issues have to be addressed by the authors:

1. When the authors introduce a specific type of NMR technique, it would be helpful for the readers not familiar with NMR analysis

to provide a 1-2 sentence description of the nature of information each specific type of NMR data (such as PRE, R1/R2 measurements, CPMG relaxation dispersion) provides. Original references to each type of NMR techniques (amide TROSY, methyl TROSY, PRE etc.) should be included in the 'Materials and Methods' section, along with the references to specific NMR experiments used for ^{15}N R1/R2 measurements.

2. Although this does not detract in any way from their accomplishments, the authors should comment on why they have chosen to use TOCSY and NOESY experiments for assignments of ILV methyls. The use of methyl out-and-back experiments (HMCMBCA and HMCBCACO) would be appropriate with the availability of backbone (CA, CB and CO) assignments and would potentially save a number of mutations.

3. The authors use R1/R2 interchangeably with T1/T2 in description of relaxation measurements in 'Materials and Methods'. This should be 'streamlined' -- e.g. in the " ^{13}C -methyl (CHD2) Relaxation Analysis" section on p. 22, it may be confusing for some readers why T2 is obtained from T1rho via R2 (and R1rho). Plus ' 15 nucleus' in the same section should read ' ^{13}C ' ?

4. Generally, 'NMR jargon' should be avoided. E.g. the readers might not know what D1 is in NMR parlance.

Reviewer #2 (Remarks to the Author):

This manuscript from Hunashal et al proposes that increased dynamics of penicillin-binding protein 5 variants (PBP5) is associated with the beta-lactam resistance of *E. faecium*. The supporting evidence is largely derived from powerful investigations of the protein in solution using NMR in which mutants associated with resistance of the organism exhibit higher dynamic behavior. As a mechanistic explanation for the resistance, it is claimed that the increased dynamics of the protein leads to increased hydrolysis of penicillin G.

The main strength of this manuscript is the application of NMR to examine the dynamic behavior of a large protein. In fact, successful assignment of ILVs for such a large protein is highly impressive and follow-up confirmation through 70 site-directed mutants indicates a high degree of rigor in the NMR work. Moreover, assigning ILVs is a clever way to get enough probes to enable measurements of relaxation dynamics. There is also plausibility to the idea that PBP5 mutants present in resistant strains exhibit increased dynamics. At the same time, there are a number of concerns that reduce the potential impact of the manuscript, including the failure to provide a clear link between specific mutants examined and resistance of *E. faecium*, the low probability that increased hydrolysis of penicillin G is a contributor to resistance, and the failure to set the work in context of what is known about other PBPs. In addition, the manuscript often fails to provide adequate interpretation of specific experiments and their significance.

Major points

The title is an overstatement on two counts. Firstly, only a single PBP is studied, and it is misleading to imply that the mechanism proposed is generally applicable to PBPs. Secondly, in the opinion of this reviewer, there is insufficient evidence presented to say unequivocally that increased dynamics is the molecular basis for resistance.

The statement in the Introduction that “no significant structural differences have been identified” in PBPs related to resistance is problematic because some published studies have revealed significant differences in structure between mutated PBPs from resistant strains and their non-mutated counterparts from susceptible strains.

The contribution of PBP5 mutations observed in resistant strains to resistance isn't clear. Statements in the second paragraph on page 9 do not offer much background. Examination of a cited reference (Rice et al 2004) showed that mutation of Met485 to Thr increases MICs for ampicillin from 38 to 48 µg/ml and to 55 µg/ml for an alanine. These numbers are not significant. The picture is similar for the serine insertion at position 466 because this elevates MICs from 38 to 46 µg/ml. The same is true for other

beta-lactams (piperacillin, ticarcillin and ceftriaxone). Hence, at least as far as can be gleaned from the manuscript and from Rice et al, neither of these mutations has a measurable impact on resistance. In the absence of a clear linkage between mutation and resistance, the conclusion from the structural studies should be qualified.

Related to this, the manuscript lacks a proper description of the T485, M485 and A485 mutations and their relationship to resistance. On Page 9, 2nd para, M485 is suddenly mentioned without introduction, leaving its significance unclear at that moment.

The conclusion that increased hydrolysis of penicillin is responsible for resistance is questionable. The hydrolysis data for M485 vs A485 in Fig. 3F do not appear significant. A more significant increase in hydrolysis is seen with the iS466 insertion mutant, but is on the orders of hours. There is no discussion of whether this timescale is physiologically relevant compared to the growth kinetics of *E. faecium*. In addition, these data should be presented as kinetic rates of deacylation (k_3) for the purposes of rigor, and for calibration with other PBPs.

Also unconvincing is the proposed linkage between increased hydrolysis and increased protein dynamics. In the background of iS466, adding A485 has only a minimal effect on hydrolysis (Fig. 4E) and yet introduction of A485 (in place of M485) increases the dynamic behavior of PBP5 (Fig. 3). This suggests hydrolysis and protein dynamics are not strongly linked.

Rather than simply provide fold changes in BOCILLIN binding for PBP5 mutants, k_2/K_s second-order rate constants should be derived as a better indicator of the significance of these data.

While the NMR data appear sound, and as noted above, highly impressive at times, interpretation for key experiments is sometimes lacking that may make it difficult for those without background in NMR to understand. As examples, it may not be clear how the chemical shift index relates to secondary structure (Figs. S3B and C). As another, the meaning of T1 and T1 rho should be explained (Fig. 3, Fig. S10). This can be accomplished easily by expansion of interpretations in the Supplemental section. Most critically, Fig. 3 also lacks sufficient interpretation for the reader to understand how panels A and B show increased dynamics.

It would also be helpful to provide more account in the Results and Materials & Methods of how the ILV backbones and sidechain methyls were assigned. The Material and Methods section lists all the triple resonance spectra collected but does not explain the procedures for making assignments.

The Conclusion is largely a recasting of the results, and while useful as an overall assessment, there is no attempt to consider the findings in light of what is known about potential resistance mechanisms in other PBPs. It needs significant expansion to address this major omission.

Specific points

The description of the protein in the parentheses at the (lines 4-5 of the first Results paragraph) is hard to follow. The meaning of “PBP5 T485” tacked on at the end is unclear at this stage, and the statement should probably be amended to say that the T485 variant protein was used for the NMR assignments rather than all structural studies.

For Fig. S12, only some T1 and T1 rho values appear to have errors. If this is because errors are very low, I would expect the circles to appear as squares. Part of the problem here is that the figure legend does not explain the error bars.

Since the whole protein has not been assigned, is it correct to assume that the CSPs in Fig. S7 E and F are for ILV residues only? This is not clear. Put another way, how can CSPs be given for each residue in the absence of full assignment?

It may not be correct to conclude that CSPs are not due to change in PBP5 conformation on the basis that crystal structures of apo vs acylated are unchanged (page 8, 2nd paragraph). Proteins may behave differently in solution vs. in a crystal lattice.

Terminology: It is not correct to call PBP5-A485 one of the “most resistant” variants (page 9, 2nd para) when talking about binding of BOCILLIN FL. Resistance relates to the strain, not a biochemical property of the protein. The same applies to mention of other serine insertions.

Corrections

Bocillin is spelled incorrectly in Fig. S2

Fig. S4 – correct “crowed” to “crowded”.

Fig. S3 – the colors for the N2 and nPB domains appear to be black and gray rather than yellow and pink.

Fig. S7 - the red dot on the CSPs (panels E and F) is presumably Ser422?

Reviewer #3 (Remarks to the Author):

This manuscript provides an elegant molecular study of major importance in the area of antibiotic mechanisms of action. PBP5 in enterococci, an essential cell wall synthesizing enzyme, is a difficult protein to purify and to study by standard biophysical methodology. The authors have succeeded in establishing an NMR technique that allows the dynamic evaluation of multiple single-step PBP5 mutants, or mutants with single amino acid insertions in the active site domain, providing insight into the mechanism of action. The study has implications for the activity of PBPs from other bacterial species and their PBP-associated resistance mechanisms.

Specific comments

1. P. 5. After the denaturing/renaturing process, does the purified PBP5 have the same enzymatic profile as wildtype enzyme, e.g., penicillin/Bocillin IC50 values in PBP labeling experiments?
2. P. 9. Were any insertion mutants generated using a residue other than serine? It would be interesting to know whether an alanine insertion would have the same effect(s).
3. P. 13. Can the authors provide some indication as to how much the various mutations affect the MICs for PenG or ceftaroline?
4. Are insertion mutations observed in other PBPs from Gram-positive pathogens? If so, this would increase the significance of the findings even more.

Minor comments

5. P. 8. Please avoid the use of generation names for cephalosporins. Ceftaroline is preferably referred to as an anti-MRSA cephalosporin.
6. The abbreviation for the non-penicillin binding domain (nPB) of PBP5 is occasionally written as “nBP”. Please check this (e.g., p. 7, line 4; Figure 1A and S1 legends).
7. There are two references numbered 36 in the reference list.

Reviewer #1 (Remarks to the Author):

I find the work by Hunashal et al. truly impressive as far as the technicalities of NMR spectroscopy are concerned. Although I cannot judge the importance or relevance of this work from biological or medical perspective, I can attest with full confidence that the presented NMR data is of superb quality considering the size (molecular weight) of the system (70 kDa). The work seems to be performed with utmost care and is well documented and illustrated. Even though no attempt was made by the authors to extract the actual parameters of dynamics (such as order parameters of N-H or C-H bond vectors or correlation times of various motions), as the material presented is of comparative nature (i.e. only the changes in relaxation parameters are tracked), I believe that the authors' conclusions are sound and would enthusiastically recommend publication of this work provided that it satisfies the requirements of Nature Communications with respect to biological/medical novelty.

We appreciate the strong support for our work by reviewer 1, especially for enthusiastically recommending publication of the work. As also highlighted in the response for reviewer 2, all amino acid substitutions in PBP5 (variants) that we systematically and thoroughly characterize arose under the selective pressure of the β -lactams used to treat infections due to *Enterococcus faecium*. This is the accepted approach in the field for identifying such resistant variants and thus this represents the most clear, direct linkage between arising mutations and resistance.

The following minor issues have to be addressed by the authors:

1. When the authors introduce a specific type of NMR technique, it would be helpful for the readers not familiar with NMR analysis to provide a 1-2 sentence description of the nature of information each specific type of NMR data (such as PRE, R1/R2 measurements, CPMG relaxation dispersion) provides. Original references to each type of NMR techniques (amide TROSY, methyl TROSY, PRE etc.) should be included in the 'Materials and Methods' section, along with the references to specific NMR experiments used for ^{15}N R1/R2 measurements.

We appreciate the suggestions of the reviewer and have implemented them in the manuscript where requested/appropriate. We have also added the appropriate citations. No specific (unique) pulse programs were used for the NMR measurements; rather, since ~2010, we generally use the manufacturer supplied pulse sequences (Bruker pulse program library) to ensure full reproducibility by any research group around the world using a modern Bruker NMR spectrometer (which nowadays is, unfortunately, nearly all groups due to the demise of Varian/Agilent). While

these pulse sequences are not always 100% optimal, the key to getting good NMR data for a system such as PBP5 is not the pulse sequences but an optimal NMR protein sample: we optimized the sample for ~24 months before initiating NMR measurements, including testing and establishing refolding, long-term stability at different temperatures, optimal buffer, among many other parameters.

2. Although this does not detract in any way from their accomplishments, the authors should comment on why they have chosen to use TOCSY and NOESY experiments for assignments of ILV methyls. The use of methyl out-and-back experiments (HMCMBCA and HMCBCACO) would be appropriate with the availability of backbone (CA, CB and CO) assignments and would potentially save a number of mutations.

HMCMBCA and HMCBCACO type experiments are certainly an additional option for the assignment of ILVs, which we have previously utilized for smaller proteins (35-45 kDa). However, with the need to assign similar residues/residue pairs within a ~600 amino acid protein, the confidence in the CA/CB differences is lower and we prefer NOE data, especially with the 3D structure being available. We always perform as many ILV mutations as needed to achieve full confidence in our ILV assignment, to ensure the highest rigor, as their interpretation is most straightforward and thus less prone to errors.

3. The authors use R1/R2 interchangeably with T1/T2 in description of relaxation measurements in 'Materials and Methods'. This should be 'streamlined' -- e.g. in the "¹³C-methyl (CHD2) Relaxation Analysis" section on p. 22, it may be confusing for some readers why T2 is obtained from T1rho via R2 (and R1rho). Plus '¹⁵ nucleus' in the same section should read '¹³C' ?

We appreciate the careful assessment by the reviewer and have unified the presentation, clarifying relationships between relaxation times and relaxation rates and updated the '¹⁵ nucleus' to ¹³C. Thank you.

4. Generally, 'NMR jargon' should be avoided. E.g. the readers might not know what D1 is in NMR parlance.

We carefully updated the manuscript to avoid NMR jargon – thank you. In general, we strive to explain each parameter/parlance upon its first mention, as we did with D1 – but we agree that it may be better to provide this information continuously, which we now do.

Reviewer #2 (Remarks to the Author):

This manuscript from Hunashal et al proposes that increased dynamics of penicillin-binding protein 5 variants (PBP5) is associated with the beta-lactam resistance of E. faecium. The supporting evidence is largely derived from powerful investigations of the protein in solution using NMR in which mutants associated with resistance of the organism exhibit higher dynamic behavior. As a mechanistic explanation for the resistance, it is claimed that the increased dynamics of the protein leads to increased hydrolysis of penicillin G.

The main strength of this manuscript is the application of NMR to examine the dynamic behavior of a large protein. In fact, successful assignment of ILVs for such a large protein is highly impressive and follow-up confirmation through 70 site-directed mutants indicates a high degree of rigor in the NMR work. Moreover, assigning ILVs is a clever way to get enough probes to enable measurements of relaxation dynamics. There is also plausibility to the idea that PBP5 mutants present in resistant strains exhibit increased dynamics. At the same time, there are a number of concerns that reduce the potential impact of the manuscript, including the failure to provide a clear link between specific mutants examined and resistance of *E. faecium*, the low probability that increased hydrolysis of penicillin G is a contributor to resistance, and the failure to set the work in context of what is known about other PBPs. In addition, the manuscript often fails to provide adequate interpretation of specific experiments and their significance.

We thank the reviewer for her/his support and highlighting the high rigor of the work and the 'clever way' of achieving a sufficiently high number of probes for rigorous relaxation dynamics experiments. We provide careful, detailed responses to all the raised concerns in the answers below.

Major points

The title is an overstatement on two counts. Firstly, only a single PBP is studied, and it is misleading to imply that the mechanism proposed is generally applicable to PBPs. Secondly, in the opinion of this reviewer, there is insufficient evidence presented to say unequivocally that increased dynamics is the molecular basis for resistance.

As requested by the reviewer we changed the title to 'The Molecular Basis for Resistance of the ESKAPE bacterium *E. faecium* Penicillin Binding Protein PBP5 to β -lactam Antibiotics'. We address the second point in the answers to the questions below.

The statement in the Introduction that "no significant structural differences have been identified" in PBPs related to resistance is problematic because some published studies have revealed significant differences in structure between mutated PBPs from resistant strains and their non-mutated counterparts from susceptible strains.

Without additional details provided by the reviewer, it is difficult to answer any specific concerns (i.e., address the specific studies referred to). However, we address the general concern below.

In our study, we observed no significant structural differences between the clinical *E. faecium* PBP5 resistance variants. Similar results have been reported for other systems. For example, the crystal structures of PBP2 from penicillin-susceptible and -resistant strains of *N. gonorrhoea* (4 mutations near the C-terminus) were within the experimental precision of the structures, i.e., identical (1). In this case, the authors concluded that the molecular basis for the reduction in acylation due to the 4 mutations was 'subtle', potentially due to either (A) slight structural changes too small to be detected by crystallography or (B) due to a dynamic equilibrium between conformational states with distinct acylation rates with increased flexibility due to the 4 mutations; however, crystallography was assumed to only trap one particular (the assumed active) state (1). The latter rationale requires dynamics between the two assumed states contributing to resistance. Similarly, an insertion mutation, D345A, in the same PBP2 of *N.*

gonorrhoea, changes the structure adjacent to the active site by extending a loop by 1 residue. However, despite this amino acid insertion, the structural differences at the active site were still negligible, being perfectly overlapped when the resistant and susceptible versions of PBP2 are superimposed (2). Whether the observed resistance was a consequence of the negative charge due to the aspartic acid, or to a change in protein dynamics was not resolved. These examples are particularly relevant because the change in resistance was due to a limited number of mutations in the PBP, like for *E. faecium* PBP5. Larger changes in structure, which can be described as 'significant', have been observed for other PBPs, such as *S. pneumoniae* PBP2x. However, in the case of PBP2x, the number of mutations between the susceptible and resistant strains in the soluble portion of PBP2x is > 80, with some immediately surrounding the active site (T338A, A347S and N514H). The PBP2x variant crystal structures show that a long loop becomes untraceable and presumably disordered in the resistant version, resulting in the rotation of a S395 sidechain away from the catalytic pocket. The authors suggest that the increased resistance is facilitated by the increased flexibility of this loop (3). Thus, despite the larger structural changes in PBP2x of *S. pneumoniae* when compared to those observed for PBP2 from *N. gonorrhoea*, the data similarly suggests that increased dynamics is likely important for conferring the observed resistance. The only technique that provides direct proof of such a flexibility, as well as reports on the timescale of these dynamics is NMR spectroscopy.

To summarize, we showed that the structural changes between the PBP5 variants (and other PBPs) are localized and small, suggesting, and as also concluded by the examples highlighted above, that dynamics likely play a role in resistance. Thus, we initiated investigations using biomolecular NMR spectroscopy, despite the large size of PBP5, which makes this a rather challenging target for rigorous NMR analysis, to understand how dynamics contributes to PBP5 function and resistance.

The contribution of PBP5 mutations observed in resistant strains to resistance isn't clear. Statements in the second paragraph on page 9 do not offer much background. Examination of a cited reference (Rice et al 2004) showed that mutation of Met485 to Thr increases MICs for ampicillin from 38 to 48 µg/ml and to 55 µg/ml for an alanine. These numbers are not significant. The picture is similar for the serine insertion at position 466 because this elevates MICs from 38 to 46 µg/ml. The same is true for other beta-lactams (piperacillin, ticarcillin and ceftriaxone). Hence, at least as far as can be gleaned from the manuscript and from Rice et al, neither of these mutations has a measurable impact on resistance. In the absence of a clear linkage between mutation and resistance, the conclusion from the structural studies should be qualified.

Let us start to answer this important concern of the reviewer by immediately highlighting that all amino acid substitutions that were characterized in this study arose under the selective pressure of the 3-lactams used to treat infections due to *Enterococcus faecium*. Thus, there is a clear and direct linkage between mutation and resistance.

Already in 1998, T. Rybkin and colleagues described 'Penicillin-binding protein 5 sequence alterations in clinical isolates of *Enterococcus faecium* with different levels of beta-lactam resistance' (4), where they came to following conclusion: "In conclusion, this study of several highly 3-lactam-resistant *E. faecium* isolates from different countries supports and extends our previous observation [9] that very high MICs of 3-lactam antibiotics are associated with low

amounts of PBP5 with decreased affinity, which relates to the presence of different amino acid substitutions, in particular at aa485 near the conserved SDN triad.”

Many concurrent studies highlighted these observations as well, which showed that, independent of research group and country of specimen collection (US, Germany, France, Sweden, Norway etc.), consistently similar substitutions and insertions were identified in PBP5 from resistant strains and, furthermore, demonstrated the importance of these PBP5 variants specifically for β -lactam resistance in *E. faecium*. They also specifically highlight the importance of substitutions of aa 485 (Met-*Thr-*Ala) and the iS466 insertion variant (4–10). As often seen in these MIC experiments, single mutations show moderate but consistent MIC changes, with multiple mutations showing significant increases – i.e., changes in aa 485 and iS466.

To highlight the strong correlation of the published resistant MIC data with our work, we now cite these publications – clearly in hindsight we should have done this before.

Lastly, a concern was raised that the publication (Rice et al 2004; (11)) showed differences in susceptibility that were not meaningful. As is common in the PBP resistance field, the impact of any single mutation in a PBP is often subtle. To overcome this (and as carefully described in the Materials and Methods section of Rice et al 2004) 10 μ g/ml arithmetic increments were used (rather than doubling dilutions) to reproducibly determine the small differences in MICs for PBP5. All experiments were performed with $n \geq 4$ and reported the arithmetic mean (as it was standard in 2004).

Taken together, there is an overwhelming amount of published data that demonstrates the importance of PBP5 in β -lactam resistance in *E. faecium*, generally, and the different variants used in this biophysical study, specifically. To further solidify this statement in the manuscript, we added a large number of citations (4–10), which confirm these statements and directly link the studied PBP5 variants with β -lactam resistance.

Related to this, the manuscript lacks a proper description of the T485, M485 and A485 mutations and their relationship to resistance. On Page 9, 2nd para, M485 is suddenly mentioned without introduction, leaving its significance unclear at that moment.

We thank the reviewer for pointing out that the overall description of the variants needs to be better explained (beyond Figure 2A). PBP5_{M485} is the least resistant variant (i.e., one could speculate wt-PBP5). PBP5_{A485} is the most resistant single mutation variant. Based on resistance, PBP5_{T485} is in the ‘middle’. This has been well-described in literature since the early 1990s, e.g. (4). We now added this description to the manuscript, referring to Figure 2A, and added additional citations to ensure full clarity.

For the NMR studies reported here, we used PBP5_{T485}, with the anticipation that chemical shift changes between the different variants are easier to track. With such a difficult NMR target one needs to decide which variant to use at an early stage – this was our ‘best guess’ when we initiated the project.

We have now expanded this section to:

.....used a BOCILLIN FL binding assay to quantify the extent of acylation in comparison to that of PBP5_{M485}, which is the least resistant PBP5 variant reported. The data show that resistant PBP5 variants exhibit an up to ~8-fold less acylation compared to PBP5_{M485} (**Fig. 2A**).

The conclusion that increased hydrolysis of penicillin is responsible for resistance is questionable. The hydrolysis data for M485 vs A485 in Fig. 3F do not appear significant. A more significant increase in hydrolysis is seen with the iS466 insertion mutant, but is on the orders of hours. There is no discussion of whether this timescale is physiologically relevant compared to the growth kinetics of *E. faecium*. In addition, these data should be presented as kinetic rates of deacylation (k₃) for the purposes of rigor, and for calibration with other PBPs.

We appreciate the suggestion by the reviewer that we should have better described the 1D ¹H NMR hydrolysis experiments. Indeed, we think that these measurements are, in fact, the most direct way to observe hydrolysis as we measure a direct change in PenG upon hydrolysis (similar to using radioactively labeled Penicillin many years ago) and not an indirect effect via Bocillin/nitrocefin binding/reaction. We performed these experiments in this manner (i.e., as described in Material and Methods) to ensure the highest reproducibility. In particular, we identified that in order to determine the hydrolysis accurately for each PBP5 variant, we had to initiate each experiment with a 10-fold (molar ratio) surplus of PenG. At the beginning of the incubation, PenG is continuously hydrolyzed by PBP5, but also continuously replaced with fresh PenG (equilibration period; PenG is still in great excess). During this equilibration period, we were able to carefully set up the NMR experiments allowing us to better identify the hydrolysis period, which shows fast hydrolysis of the PenG-PBP5 acylenzyme when it can no longer be replaced by unhydrolyzed PenG (as it occurs during the equilibration period). This is described in detail in the Material and Methods section.

We also tested these experiments with 1:1 and 2:1 ratios of PenG:PBP5, but NMR-based limitations as well as the speed of hydrolysis only enabled us to 'catch' the reaction already in progress. Thus, while we could have tried 5:1 etc., we decided to use a 10:1 PenG:PBP5 ratio, which enabled us to set up the experiment in the most careful way possible (especially for the iSer466 variants). All experiment were consistently performed at 35°C, close to physiological conditions (*E. faecium* optimal growth condition are between 42-45°C), but ≥20°C below the PBP5 melting temperature, ensuring long term stability of the enzyme, as confirmed by our high-quality data of PBP5 that did not change over a period of 6+ weeks in the NMR spectrometer.

PBP5_{A485} reached the hydrolysis endpoint ~20 hrs quicker than PBP5_{M485}; about 25% faster when considering the total measurement time. PBP5_{iS466} reached the hydrolysis endpoint ~70% quicker than PBP5_{M485}. These are all clear endpoint changes. Upon the onset of full hydrolysis (see above), the full hydrolysis occurs swiftly in a matter of minutes to hours.

We prefer to look at hydrolysis endpoints than deacylation (k₃) rates as this analysis can be readily performed for any PBP using the setup described in the methods and thus are fully comparable. Importantly, these are simple ¹H 1D NMR measurements that require small amounts of protein and report on direct changes of PenG.

Also unconvincing is the proposed linkage between increased hydrolysis and increased protein dynamics. In the background of iS466, adding A485 has only a minimal effect on hydrolysis (Fig. 4E) and yet introduction of A485 (in place of M485) increases the dynamic behavior of PBP5 (Fig. 3). This suggests hydrolysis and protein dynamics are not strongly linked.

We apologize that we failed to make the connection between increased hydrolysis and increased protein dynamics clearer. Indeed, it is quite striking that the different PBP5 variants display a different dynamic behavior, and that this behavior also changes the 3-lactam hydrolysis. In particular, the data show that reducing the bulkiness of the PBP5 485 amino acid side chain (from M-*T-*A) results in increased local dynamics of PBP5, which happens independently of 3-lactam/penicillin binding. As highlighted in Figure 5, the reduction in the 485-sidechain bulkiness leads to a novel internal cavity (close to the active site residue S422) and increased local dynamics, as assayed via ¹³C ILV sidechain dynamics in our study. This increased dynamics likely leads to increased water access and thus hydrolysis, as again highlighted by our NMR data.

Indeed, to ensure that this fast timescale dynamics (increased local dynamics) does not lead to an undetected change in intermediate timescale dynamics and thus PBP5 function, we recorded ¹³C ILV ct-CPMG data that directly reports on intermediate timescale dynamics (reported in Figure S14). This is the most direct and rigorous way to directly show this type of dynamics. By excluding intermediate timescale dynamics in this observed event, the fast timescale dynamics is the sole effect that drives the increased PenG hydrolysis that we detect in our direct 1D ¹H NMR measurements of PenG hydrolysis (i.e., we measure the change directly on the 3-lactam/penicillin). We are not aware of how we could have shown this correlation more directly with the same high level of rigor, than with the approach used in this study.

The reviewer is correct that the effect is different for the clinical lid insertion variant iS466. Indeed, iS466 does not show increased dynamics vs PBP5_{M485} in its free form. However, the complex of PBP5_{iSer466}-PenG shows much increased side chain dynamics. The lid is a structural element that cradles the top of the 3-lactam binding cleft. As highlighted also in our response to reviewer 3, we made multiple non-clinically observed insertions of serine residues at positions 467, 468, 469, 470 and 471 (i.e., testing most of the lid residues that cradle the 3-lactam binding cleft). None of them showed the same effect as the clinical variant iS466 (indeed they showed less resistance than PBP5_{T485}). These data are shown in Figure 2A. Thus, we have established that the position of the insertion within the lid is important for the function of iS466. Again, the increased local lid dynamics most likely allows for increased water access to the PBP5 active site, allowing for increased hydrolysis of the 3-lactam/PenG). Again, ¹³C ILV ct-CPMG data recorded on PBP5_{iSer466}-PenG does not show intermediate timescale dynamics for residues that gain fast timescale dynamics.

Rather than simply provide fold changes in BOCILLIN binding for PBP5 mutants, k₂/K_s second-order rate constants should be derived as a better indicator of the significance of these data.

As detailed above, all clinical variants of PBP5 have been reported and the main reasons we performed the BOCILLIN binding experiments are 2-fold: First, we wanted to show that using this method the previously reported resistance ranking does not change and our data fully agrees with previously published work (carefully reproducing and expanding the set of data). It does. Second,

we wanted to establish an assay to test non-clinical variants of PBP5. e.g., we created many additional insertion lid variants, including iS467, iS468, iS469, iS470, iS471. Our data show that only insertion at position iS466 shows a large effect on BOCILLIN binding.

This work was not designed as a detailed BOCILLIN biochemistry study, but rather to investigate the biophysical basis for the observed differences in resistance in these well-established clinical variants using X-ray crystallography, NMR spectroscopy and NMR-based hydrolysis assays. These comprehensive biophysical experiments allowed us to fully define the changes in structure (none) and dynamics (many) of these PBP5 resistance variants.

While the NMR data appear sound, and as noted above, highly impressive at times, interpretation for key experiments is sometimes lacking that may make it difficult for those without background in NMR to understand. As examples, it may not be clear how the chemical shift index relates to secondary structure (Figs. S3B and C). As another, the meaning of T₁ and T₁ rho should be explained (Fig. 3, Fig. S10). This can be accomplished easily by expansion of interpretations in the Supplemental section. Most critically, Fig. 3 also lacks sufficient interpretation for the reader to understand how panels A and B show increased dynamics.

We thank the reviewer for pointing out that more details for our NMR data will be useful for the readership. The chemical shift index compares the experimental chemical shifts, which are sensitive to the α / ϕ backbone angles of PBP5 (here we used C α and C13 chemical shift, which are most commonly used), to random coil chemical shifts (here we used the RefDB database (12)); positive or negative deviations correlate with α -helical or 13-sheet secondary structure. As suggested, this information has been added to Figs. S3B/C.

As requested, we added a description of T₁ and T_{1n}. T₁ is longitudinal relaxation, which describes the recovery time of the magnetization to the external magnetic field B₀. T_{1n} describes rotating frame relaxation, i.e., relaxation along the radio frequency field depending on an applied spin-lock field. T_{1n} is slower than T₂ and thus easier to measure in ¹³C ILV experiments, but together with the T₁ experiment can be translated into T₂ times if needed.

As requested, we added "Residues with increased dynamics have negative delta T_{1ρ} values and are annotated." to the figure caption of Fig. 3 to allow for direct interpretation of the data.

It would also be helpful to provide more account in the Results and Materials & Methods of how the ILV backbones and sidechain methyls were assigned. The Material and Methods section lists all the triple resonance spectra collected but does not explain the procedures for making assignments.

We refer to all software programs used for the assignment of PBP5. Assigning a protein of the size of PBP5 is not much different than a small protein, only that bookkeeping becomes critical (using Cara is very helpful and our 'go-to' program) and many single amino acid labeling schemes/samples, un-labeling scheme samples and a lot of NMR time. Dr. Senthil Ganesan, who performed the backbone assignment, has incredibly extensive experience in assigning large proteins, which was certainly critical. Lastly, we also assigned the PenG bound state – this ensures some overlapped peaks became clear. Regarding the ILV assignment, we initiated the

assignment using a ^{15}N and a $^{15}\text{N}/^{13}\text{C}$ filtered NOE spectrum to correlate the backbone assignments with the obvious ILV assignments. We then proceeded with $^{13}\text{C}/^{13}\text{C}$ filtered NOE data as well as many single point variants of PBP5 to fully assign the ILVs in PBP5, which was achieved by Dr. Yamanappa Hunashal. No tricks, just careful hard work, that required many samples, many days of NMR time and two very smart, dedicated and experienced co-workers to get the job done.

We used 600 and 800 MHz NMR spectrometers: Bruker Neo with modern TCI HCN active z-gradient cryo-probes. Critically, we maintain our spectrometers meticulously and calibrate them continuously to achieve the highest S/N etc.

The Conclusion is largely a recasting of the results, and while useful as an overall assessment, there is no attempt to consider the findings in light of what is known about potential resistance mechanisms in other PBPs. It needs significant expansion to address this major omission.

We agree with the reviewer that it is important to assess our findings and compare them with the available data in the discussion. But we do not agree that detailing our results in the discussion is simply a 'recasting' of the data, since our work/data described in this manuscript addresses, for the first-time, flexibility/dynamics in a high-molecular weight PBP, the well-recognized essential target of β -lactam antibiotics. As such, we focused our discussion on the novelty of the results obtained by NMR spectroscopy to describe and quantify PBP5 dynamics, rather than provide an extensive discussion of previous models based on kinetic analyses and X-ray structures.

That said, in response to the reviewer, we have expanded our discussion to explain these points more fully. As summarized above, multiple crystallography-based studies have highlighted the potential role of increased flexibility in the interaction of PBPs (and their resistant variants) with β -lactams and the stability of the acylenzyme (often due to simply the exclusion of all other possibilities, as the data did not show any other rational explanation of how resistance was achieved—as said, only NMR spectroscopy can provide a direct proof of protein dynamics). At the time, using biomolecular solution-state NMR spectroscopy to probe these systems was not attempted due to the large size of high-molecular weight PBPs, such as PBP5 (>60 kDa). Only very recently, an elegant study by Dr. Pei Zhou (Duke University) aimed to overcome this issue and to address the question of PBP protein dynamics using NMR spectroscopy. In his work, ^{19}F NMR spectroscopy was leveraged to study PBP2 from *Neisseria gonorrhoeae* by following bromotrifluoroacetone modification of a Cys residue introduced by site-directed mutagenesis (13). The advantage of this approach is that it overcomes the need for the sequence-specific backbone assignment and ILV methyl side-chain assignment of the PBP under investigation, which is very time consuming, difficult and expensive. The (significant) drawback is the much more indirect nature of the approach and the limited number of reporters, which might not report the full mechanism. While insightful, it lacked the full description that we provide here for PBP5. Here, we directly report on the flexibility and dynamics in this protein, and its mutants, via hundreds of reporters distributed throughout the protein and, for PBP5, provide the first molecular insights into the mechanism of PBP5 resistance in the clinic. These points have now been added to the discussion.

Specific points

The description of the protein in the parentheses at the (lines 4-5 of the first Results paragraph) is hard to follow. The meaning of “PBP5 T485” tacked on at the end is unclear at this stage, and the statement should probably be amended to say that the T485 variant protein was used for the NMR assignments rather than all structural studies.

We thank the reviewer for pointing this out and we have updated the manuscript accordingly.

PBP5₃₇₋₆₇₈ (soluble PBP5; 641 aa; 69.7 kDa; $T_m = 333$ K; for all NMR assignment studies PBP5 T485 was used) is a large protein for biomolecular NMR spectroscopy.

For Fig. S12, only some T1 and T1 rho values appear to have errors. If this is because errors are very low, I would expect the circles to appear as squares. Part of the problem here is that the figure legend does not explain the error bars.

We have updated the figure captions to explain the error bars. All reported values have error bars – furthermore, all data is fully available on figshare to allow for full data reproducibility.

Since the whole protein has not been assigned, is it correct to assume that the CSPs in Fig. S7 E and F are for ILV residues only? This is not clear. Put another way, how can CSPs be given for each residue in the absence of full assignment?

Figure captions for Figs. S7E/F highlight that these CSI are based on ^{15}N chemical shifts changes from the 2D [^1H , ^{15}N] TROSY spectra shown in Fig. S7A and C.

It may not be correct to conclude that CSPs are not due to change in PBP5 conformation on the basis that crystal structures of apo vs acylated are unchanged (page 8, 2nd paragraph). Proteins may behave differently in solution vs. in a crystal lattice.

Over the years it has been shown that the 3D structures derived via different techniques, such as X-ray crystallography, NMR spectroscopy and cryo-EM agree with each other and furthermore that data derived from structural data agrees with cellular derived data (as shown via mutagenesis as well as simply by modern structure-based drug design). Large for-profit corporations would not invest billions of dollars into structure-based drug design if this correlation was incorrect.

Lastly, many research groups have established that the difference between crystal structures and in solution NMR data that leads to changes in biological data interpretation is due to protein dynamics, which is key to the function of proteins. The sole technique that allows for the measurement of dynamics is indeed NMR spectroscopy. Due to its more difficult nature – from sample preparation, data collection and data evaluation, it is less often used than X-ray crystallography or cryo-EM. Nevertheless, only NMR spectroscopy can report on dynamics at different timescales. Unfortunately, NMR spectroscopy is not the most user-friendly approach and requires experts with much experience (as well it is very costly in regard to sample preparation, measurement time etc) and thus few reports about protein dynamics are published, despite its direct importance with the biological function and events.

Terminology: It is not correct to call PBP5-A485 one of the “most resistant” variants (page 9, 2nd para) when talking about binding of BOCILLIN FL. Resistance relates to the strain,

not a biochemical property of the protein. The same applies to mention of other serine insertions.

We updated as requested.

Corrections

Bocillin is spelled incorrectly in Fig. S2

Thank you – updated.

Fig. S4 – correct “crowed” to “crowded”.

Thank you – updated.

Fig. S3 – the colors for the N2 and nPB domains appear to be black and gray rather than yellow and pink.

Thank you – updated.

Fig. S7 - the red dot on the CSPs (panels E and F) is presumably Ser422?

Thank you – updated.

Reviewer #3 (Remarks to the Author):

This manuscript provides an elegant molecular study of major importance in the area of antibiotic mechanisms of action. PBP5 in enterococci, an essential cell wall synthesizing enzyme, is a difficult protein to purify and to study by standard biophysical methodology. The authors have succeeded in establishing an NMR technique that allows the dynamic evaluation of multiple single-step PBP5 mutants, or mutants with single amino acid insertions in the active site domain, providing insight into the mechanism of action. The study has implications for the activity of PBPs from other bacterial species and their PBP-associated resistance mechanisms.

We appreciate the strong support of reviewer 3, highlighting that this is an ‘elegant molecular study of major importance in the area of antibiotic mechanisms of action’ and that ‘The study has implications for the activity of PBPs from other bacterial species and their PBP-associated resistance mechanisms’.

Specific comments

1. P. 5. After the denaturing/renaturing process, does the purified PBP5 have the same enzymatic profile as wildtype enzyme, e.g., penicillin/Bocillin IC50 values in PBP labeling experiments?

We have tested the refolded protein in multiple ways, including by showing an identical melting temperature (T_m), crystalizing the refolded protein (structure reported in the manuscript), and

identical NMR spectrum (minus the not exchanged H^N/N peaks). Indeed, activity assays for PBP5 are difficult as these enzymes have very low turnover as well as its optimal substrates for in vitro studies are ill-defined (which peptide-glycan precursor etc.). It is for this matter that Bocillin or nitrocefin binding assay are used to indirectly show PBP5 activity.

2. P. 9. Were any insertion mutants generated using a residue other than serine? It would be interesting to know whether an alanine insertion would have the same effect(s).

We agree with the reviewer that it is indeed an interesting question to determine if there is anything 'special' regarding a serine being inserted at this position.

Literature data have reported clinical variants of PBP5 that have an ASP amino acid inserted at position 466 in lieu of a SER. Both Ser and Asp have a similar length; furthermore, this points to the importance of having the ability to form a hydrogen bond, which might be important for the water transport for the increased hydrolysis. We have not tested if a different aa can replace Ser or Asp and no different clinical variant has been reported so far.

Furthermore, we made multiple non clinical insertions of serine residues at position 467, 468, 469, 470 and 471 to test most of the lid residues that surround the β -lactam binding cleft, but none of them showed the same effect as the clinical variant S466 (indeed they showed higher resistance than T485). This data is shown in Figure 2. Thus, we have also established that the position within the lid is important.

3. P. 13. Can the authors provide some indication as to how much the various mutations affect the MICs for PenG or ceftaroline?

We do not have specific information regarding the activity of ceftaroline or penicillin against these mutants, but some indication can be gained from Rice et al (11). In general, ceftaroline has poor activity against *E. faecium* strains (14, 15), and its activity tracks with Ceftriaxone. In Rice et al, ceftriaxone MICs ranged from ~2000 pg/ml for wild type PBP to 6,300 pg/ml for the mutant with all four aa changes. Penicillin, on the other hand, generally tracks with piperacillin in being somewhat less active than ampicillin. Piperacillin MICs in the prior studies ranged from 200 pg/ml for wild type to 600 pg/ml for the mutant with all four aa changes.

4. Are insertion mutations observed in other PBPs from Gram-positive pathogens? If so, this would increase the significance of the findings even more.

Resistance in many naturally transformable Gram-positive (*S. pneumoniae*, alpha streptococci) and Gram-negative (*Neisseria*) species commonly occurs through the importation of segments of other PBP, forming mosaic PBPs. Ceftobiprole resistance has been associated with a 5 amino acid insertion in the non-catalytic end of PBP2a (16). Single amino acid insertions near the active site have been associated with resistance in several Gram-negative species – *Helicobacter* (17); *Haemophilus* (18); *N. gonorrhoeae* (2, 19).

Minor comments

5.P. 8. Please avoid the use of generation names for cephalosporins. Ceftaroline is preferably referred to as an anti-MRSA cephalosporin.

We thank the reviewer for pointing this out. We have added the information into the manuscript.

6. The abbreviation for the non-penicillin binding domain (nPB) of PBP5 is occasionally written as “nBP”. Please check this (e.g., p. 7, line 4; Figure 1A and S1 legends).

Updated throughout the manuscript – thank you.

7. There are two references numbered 36 in the reference list.

Updated – thank you.

References

1. Powell, A. J., Tomberg, J., Deacon, A. M., Nicholas, R. A., and Davies, C. (2009) Crystal structures of penicillin-binding protein 2 from penicillin-susceptible and -resistant strains of *Neisseria gonorrhoeae* reveal an unexpectedly subtle mechanism for antibiotic resistance. *J. Biol. Chem.* **284**, 1202–1212
2. Fedarovich, A., Cook, E., Tomberg, J., Nicholas, R. A., and Davies, C. (2014) Structural effect of the Asp345a insertion in penicillin-binding protein 2 from penicillin-resistant strains of *Neisseria gonorrhoeae*. *Biochemistry*. **53**, 7596–7603
3. Dessen, A., Mouz, N., Gordon, E., Hopkins, J., and Dideberg, O. (2001) Crystal structure of PBP2x from a highly penicillin-resistant *Streptococcus pneumoniae* clinical isolate: a mosaic framework containing 83 mutations. *J Biol Chem.* **276**, 45106–45112
4. Rybkine, T., Mainardi, J. L., Sougakoff, W., Collatz, E., and Gutmann, L. (1998) Penicillin-binding protein 5 sequence alterations in clinical isolates of *Enterococcus faecium* with different levels of beta-lactam resistance. *J. Infect. Dis.* **178**, 159–163
5. Galloway-Peña, J. R., Rice, L. B., and Murray, B. E. (2011) Analysis of PBP5 of early U.S. isolates of *Enterococcus faecium*: sequence variation alone does not explain increasing ampicillin resistance over time. *Antimicrob. Agents Chemother.* **55**, 3272–3277
6. Jureen, R., Mohn, S. C., Harthug, S., Haarr, L., and Langeland, N. (2004) Role of penicillin-binding protein 5 C-terminal amino acid substitutions in conferring ampicillin resistance in Norwegian clinical strains of *Enterococcus faecium*. *APMIS*. **112**, 291–298
7. Huycke, M. M., Sahm, D. F., and Gilmore, M. S. (1998) Multiple-drug resistant enterococci: the nature of the problem and an agenda for the future. *Emerg Infect Dis.* **4**, 239–249
8. Torell, E., Cars, O., and Hambraeus, A. (2001) Ampicillin-resistant enterococci in a Swedish university hospital: nosocomial spread and risk factors for infection. *Scand J Infect Dis.* **33**, 182–187
9. Ligozzi, M., Pittaluga, F., and Fontana, R. (1996) Modification of penicillin-binding protein 5 associated with high-level ampicillin resistance in *Enterococcus faecium*. *Antimicrob Agents Chemother.* **40**, 354–357
10. Poeta, P., Costa, D., Igrejas, G., Sáenz, Y., Zarazaga, M., Rodrigues, J., and Torres, C. (2007) Polymorphisms of the *pbp5* gene and correlation with ampicillin resistance in *Enterococcus faecium* isolates of animal origin. *J Med Microbiol.* **56**, 236–240
11. Rice, L. B., Bellais, S., Carias, L. L., Hutton-Thomas, R., Bonomo, R. A., Caspers, P., Page, M. G. P., and Gutmann, L. (2004) Impact of specific *pbp5* mutations on expression of beta-lactam resistance in *Enterococcus faecium*. *Antimicrob. Agents Chemother.* **48**, 3028–3032
12. Zhang, H., Neal, S., and Wishart, D. S. (2003) RefDB: a database of uniformly referenced protein chemical shifts. *J Biomol NMR.* **25**, 173–195
13. Fenton, B. A., Tomberg, J., Sciandra, C. A., Nicholas, R. A., Davies, C., and Zhou, P. (2021) Mutations in PBP2 from ceftriaxone-resistant *Neisseria gonorrhoeae* alter the dynamics of the 133-134 loop to favor a low-affinity drug-binding state. *J Biol Chem.* **297**, 101188
14. Ge, Y., Biek, D., Talbot, G. H., and Sahm, D. F. (2008) In vitro profiling of ceftaroline against a collection of recent bacterial clinical isolates from across the United States. *Antimicrob Agents Chemother.* **52**, 3398–3407
15. Karlowsky, J. A., Adam, H. J., Decorby, M. R., Lagacé-Wiens, P. R. S., Hoban, D. J., and Zhanel, G. G. (2011) In vitro activity of ceftaroline against gram-positive and gram-negative pathogens isolated from patients in Canadian hospitals in 2009. *Antimicrob Agents Chemother.* **55**, 2837–2846
16. Morroni, G., Brenciani, A., Brescini, L., Fioriti, S., Simoni, S., Pocognoli, A., Mingoia, M., Giovanetti, E., Barchiesi, F., Giacometti, A., and Cirioni, O. (2018) High Rate of Ceftobiprole

- Resistance among Clinical Methicillin-Resistant *Staphylococcus aureus* Isolates from a Hospital in Central Italy. *Antimicrob Agents Chemother.* **62**, e01663-18
17. Okamoto, T., Yoshiyama, H., Nakazawa, T., Park, I.-D., Chang, M.-W., Yanai, H., Okita, K., and Shirai, M. (2002) A change in PBP1 is involved in amoxicillin resistance of clinical isolates of *Helicobacter pylori*. *J Antimicrob Chemother.* **50**, 849–856
 18. Kitaoka, K., Kimura, K., Kitanaka, H., Banno, H., Jin, W., Wachino, J.-I., and Arakawa, Y. (2018) Carbapenem-Nonsusceptible *Haemophilus influenzae* with Penicillin-Binding Protein 3 Containing an Amino Acid Insertion. *Antimicrob Agents Chemother.* **62**, e00671-18
 19. Brannigan, J. A., Tirodimos, I. A., Zhang, Q. Y., Dowson, C. G., and Spratt, B. G. (1990) Insertion of an extra amino acid is the main cause of the low affinity of penicillin-binding protein 2 in penicillin-resistant strains of *Neisseria gonorrhoeae*. *Mol Microbiol.* **4**, 913–919

REVIEWERS' COMMENTS

Reviewer #2 (Remarks to the Author):

The manuscript has been strengthened in terms of conveying the role of specific PBP5 mutations in the resistance of *E. faecium* and makes better reference to the literature.

However, the authors appear to have missed the point regarding experiments showing hydrolysis of penicillin G. The concern wasn't what was being measured, or how it was being measured, but rather whether it has biological significance. Is the ~20 hrs difference in endpoints between M485 vs A485 meaningful in physiological terms. It is not until around 50 hours that any real difference in levels of hydrolysis is seen and even then is only about 10% different. As noted in the first review, the iS466 mutant does hydrolyze penicillin G faster but the timescale is still hours rather than minutes. In the rebuttal, the authors have not addressed the concern as to whether this timescale is significant in terms of growth rates of *E. faecium*. This remains a significant point that goes to the heart of the biological claims made.

Reviewer #3 (Remarks to the Author):

The authors have conscientiously addressed my previous concerns. The paper will be a sound contribution to the area of PBP-mediated resistance in *E. faecium*.

Reviewer #2 (Remarks to the Author):

The manuscript has been strengthened in terms of conveying the role of specific PBP5 mutations in the resistance of *E. faecium* and makes better reference to the literature.

However, the authors appear to have missed the point regarding experiments showing hydrolysis of penicillin G. The concern wasn't what was being measured, or how it as being measured, but rather whether it has biological significance. Is the ~20 hrs difference in endpoints between M485 vs A485 meaningful in physiological terms. It is not until around 50 hours that any real difference in levels of hydrolysis is seen and even then is only about 10% different. As noted in the first review, the iS466 mutant does hydrolyze penicillin G faster but the timescale is still hours rather than minutes. In the rebuttal, the authors have not addressed the concern as to whether this timescale is significant in terms of growth rates of *E. faecium*. This remains a significant point that goes to the heart of the biological claims made.

Our careful NMR studies clearly established that clinically relevant amino acid substitutions in PBP5 increase the rate of hydrolysis of the acylenzyme thereby directly contributing to a decrease in the occupancy of the active site by penicillin. The fact that this reaction contributes to resistance does not require any further demonstration. The reviewer questions the fact that the amplitude of the observed hydrolysis reaction may not be sufficient to account for resistance. There is no experimental approach that can rigorously address this question since the minimal inhibitory concentration is not a numeric function of a single kinetic parameter such as the deacylation rate.

Indeed, the acylation of PBP5 by penicillin in the periplasm occurs in competition with the binding of the peptidoglycan precursors to PBP5, both of which are anchored in the outer leaflet of the cytoplasmic membrane. Binding of penicillin to PBP5 is also competitive with respect to the acylation of high-affinity PBPs by penicillin. Another difficulty arises from the conditions of the assay. NMR studies are performed in conditions that have been optimized for data collection. These conditions do not 100% reflect the conditions prevailing in the cytoplasm (ionic strength, osmolarity, binding of PBP5 to partner enzymes in the peptidoglycan polymerization complexes). Those conditions are mostly not compatible with high quality, rigorous and especially reproducible NMR data collection. We thus strongly advocate that further discussion on the extent of the increase in the hydrolysis of the acyl enzyme would, by essence, be highly speculative and not particularly informative.

Reviewer #3 (Remarks to the Author):

The authors have conscientiously addressed my previous concerns. The paper will be a sound contribution to the area of PBP-mediated resistance in *E. faecium*.

We appreciate the support of reviewer 3.